# RETHINKING ADVERSARIAL ROBUSTNESS IN THE CONTEXT OF THE RIGHT TO BE FORGOTTEN

## ABSTRACT

The past few years have seen an intense research interest in the practical needs of the "right to be forgotten", which enables machine learning models to unlearn a fraction of training data and its lineage. As a result of this growing interest, numerous *machine unlearning* methods have been proposed and developed to address this important aspect of data privacy. While existing machine unlearning methods prioritize the protection of individuals' private and sensitive data, they overlook investigating the unlearned models' susceptibility to adversarial attacks and security breaches. In this work, we uncover a novel security vulnerability of machine unlearning based on the insight that the adversarial vulnerabilities can be bolstered especially for adversarial robust models. To exploit this observed vulnerability, we propose a novel attack called *Adversarial Unlearning Attack* (AdvUA), which aims to generate a small fraction of malicious unlearning requests during the unlearning process. AdvUA causes a significant reduction of adversarial robustness in the unlearned model compared to the original model, providing an entirely new capability for adversaries that is infeasible in conventional machine learning pipelines. Notably, we also show that AdvUA can effectively enhance model stealing attacks by extracting additional decision boundary information, further emphasizing the breadth and significance of our research. Extensive numerical studies are conducted to demonstrate the effectiveness of the proposed attack. *Our code is available in the supplementary material.*

## 1 INTRODUCTION

In recent years, many countries have raised concerns about protecting personal privacy. In practice, users may choose to have their data completely removed from a system, especially sensitive systems such as those do with finance or healthcare (Nguyen et al., 2022). Recent regulations (e.g., the well-known European Union's GDPR (Voigt & Von dem Bussche, 2017)) now compel organizations to give users "the right to be forgotten", i.e., the right to have all or part of their data deleted from a system on request (Zhang et al., 2022b; Graves et al., 2021; Chen et al., 2021). The most straightforward approach is to retrain the model on all data except the portion that has been removed, but this approach is in general impractical since the computational resources consumed are usually costly. Thus, aiming to efficiently remove data as well as their contribution to the model, a new machine learning privacy protection research direction has emerged, called *machine unlearning*.

Numerous research efforts have been dedicated to addressing the challenge of data removal in inefficient retraining. Two prominent research fields that have emerged in this context are *exact unlearning* (Yan et al., 2022; Yu et al., 2022; Bourtoule et al., 2021b) and *approximate unlearning* (Chen et al., 2022; Gupta et al., 2021; Neel et al., 2021; Sekhari et al., 2021). Exact unlearning aims to completely reverse the effects of the previously learned data points. Instead of aiming for a perfect reversal of the learned data, approximate unlearning seeks to achieve a reasonably close approximation. This can be beneficial in situations where complete unlearning is computationally expensive or impractical, providing a compromise between removal efficiency and performance retention.

However, current machine unlearning methods exhibit an important limitation as they primarily concentrate on efficiently removing the effects of specific data instances from a trained machine learning model. Consequently, it remains uncertain whether these existing techniques might unexpectedly influence the adversarial robustness of the associated machine learning models. Note that adversarial

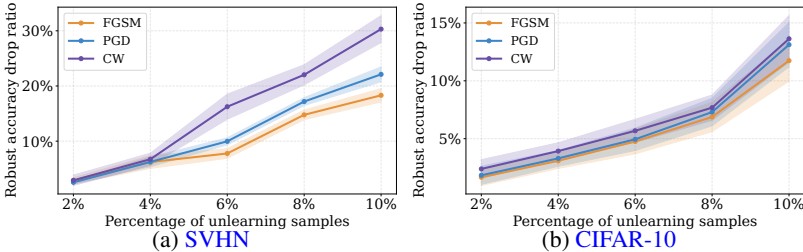

Figure 1: Robustness degradation of defended robust models against FGSM, PGD, and CW attacks after randomly removing different percentages of training samples on SVHN and CIFAR-10.

robustness, in the context of machine learning, refers to the ability of a trained model to maintain its accuracy and performance even when it is exposed to deliberately perturbed input data known as adversarial examples (Li et al., 2019; Goodfellow et al., 2014; Szegedy et al., 2013). Despite the great importance of studying adversarial robustness from the selective forgetting perspective, there is no existing work exploring the adversarial robustness properties of the unlearned models in the context of the right to be forgotten. Therefore, it is natural to ask

*Q1: Does unlearning amplify the vulnerability of unlearned models to adversarial attacks?*

To answer this question, we first conducted initial experiments to investigate adversarial robustness degradation of the defended and undefended models[1] by randomly deleting some training samples in the context of machine unlearning. In the experiments, the defended robust models are constructed using adversarial training (Madry et al., 2018; Goodfellow et al., 2014), which proves to be the most effective method against adversarial attacks (Bai et al., 2021; Pang et al., 2020; Maini et al., 2020). The experimental results are presented in Fig. 1, with additional details available in Appendix E. Notably, these experimental results show that the random removal strategy substantially amplifies the adversarial vulnerabilities of defended robust models. Additionally, in our experiments, we also found that compared with naturally undefended models, adversarially robust models are indeed *more susceptible* to malicious unlearning samples. The reason is that existing robust training methods heavily rely on the training data to enhance the model's robustness against adversarial attacks. Motivated by this, it is important to further investigate that

*Q2: Are there specific requested unlearning samples that play a pivotal role in generating these new successful adversarial examples, which were unattainable before unlearning?*

This work aims to provide answers to the above questions and highlight a potential adversarial attack vulnerability in the unlearning process - an adversary can make use of the unlearning pipeline to craft malicious unlearning requests to achieve his desired adversarial attack goals. Note that like traditional data poisoning attacks, the recent works (Hu et al., 2023; Qian et al., 2023; Di et al., 2022) still focus on how to poison the training data, and fail to study the impact of unlearning on the models' vulnerabilities to adversarial attacks. Therefore, in this paper, we aim to conduct an investigation into the adversarial risks associated with exercising the "right to be forgotten" from machine learning models, without altering the training data.

**Our Contributions.** To develop an understanding of adversarial risks associated with the process of unlearning, we in this paper present a novel attack named ***Adversarial Unlearning Attack*** (***AdvUA***), which exploits the unlearning pipeline to increase adversarial vulnerabilities. The key idea behind AdvUA is to select unlearning samples that are not only in close proximity to the target victim samples but also align with the adversarial attack directions. Specifically, in our proposed method, we first design a distance evaluation metric to estimate the space-filling capability of the region surrounding the target victim samples in the representation space. Then, based on the insight that not all nearest neighbor samples are equally critical for performing adversarial unlearning attacks, we propose a direction alignment loss to closely match the adversarial attack with the unlearning attack. *Our method is orthogonal to existing approaches on adversarial attacks and can be easily integrated with them to create advanced adversarial attack strategies.* Furthermore, it is worth highlighting that AdvUA can also *bolster the effectiveness of model stealing attacks* by extracting more decision boundary information, underlining the extensive scope and importance of our research.

---

[1]Throughout the paper, we use "undefended (natural) model" and "defended (robust) model" to denote the machine learning model with natural training algorithm and robust training algorithm, respectively.

We also empirically illustrate that AdvUA achieves a high attack success rate on various benchmarks, including CIFAR-10 (Krizhevsky & Hinton, 2010) and ImageNet (Deng et al., 2009), against various robust learning methods in Section 4. Our evaluation also indicates that AdvUA performs well across different model architectures and machine unlearning methods. Overall, by conducting this study, we aim to shed light on the potential consequences of applying machine unlearning techniques to adversarially robust models and to gain insights into the interplay between data removal and model robustness against adversarial attacks. Ultimately, our findings will contribute to a more comprehensive understanding of the implications of machine unlearning in the context of adversarial machine learning and its implications for real-world applications.

## 2 BACKGROUND AND RELATED WORK

**Notations and Machine Unlearning.** Let $S = \{(x_i, y_i)\}_{i=1}^n$ denote the dataset, where $x_i \in \mathcal{X} \subset \mathbb{R}^d$ is a $d$-dimensional feature and $y_i \in \mathcal{Y} = \{1, \cdots, C\}$. Let us suppose that a $C$-label classifier $F(W) : \mathbb{R}^d \to \mathbb{R}^C$ labels a sample $x$ as $\arg\max_{c \in \mathcal{Y}} F(x; W)[c]$, where $W \in \mathcal{W}$ represents the parameters of $F$. For $F$, we denote $H$ as the representation learning function and $G$ as the final prediction head, i.e., $F(W) = G \circ H$. Given the learning algorithm $L$ and $S$, the model owner can train a model $F(W^*)$ such that $F(W^*)$ achieves a low empirical loss. In machine unlearning, users can submit data removal requests $S_u \subset S$ to eliminate the influence of $S_u$ from $W^*$, leading to the creation of the unlearned model $W^u \in \mathcal{W}$. Note that machine unlearning can be divided into: *exact* and *approximate*. Below, we outline the definitions of existing approximate and exact unlearning techniques (Warnecke et al., 2023; Gupta et al., 2021; Neel et al., 2021; Guo et al., 2019).

**Definition 1.** *Consider a learning algorithm $L$ and its unlearning function $U_L : (S, L(S), S_u) \to W^u$, where $W^u$ is the unlearned model. Note that the original model $W^*$ is derived from $S$ and $L$, and the goal of $U_L$ is to modify this original model to unlearn the requested forget set $S_u$. The pair $(L, U_L)$ achieves exact unlearning if $\forall S$ and $S_u \subset S, \Pr(L(S_r)) = \Pr(U_L(S, L(S), S_u))$, where $S_r = S \setminus S_u$. This implies that it becomes indistinguishable whether the model was trained after unlearning $S_u$ from $L(S)$ or if it was trained exclusively on $S_r$. The pair $(L, U_L)$ satisfies $(\gamma, \zeta)$-unlearning if $\forall S, S_u \subset S$, and $\forall \mathcal{Z} \subseteq \mathcal{W}, \Pr(U_L(S, L(S), S_u) \in \mathcal{Z}) \leq e^\gamma \Pr(L(S_r) \in \mathcal{Z}) + \zeta$ and $\Pr(L(S_r) \in \mathcal{Z}) \leq e^\gamma \Pr(U_L(S, L(S), S_u) \in \mathcal{Z}) + \zeta$.*

**Related Work.** Since the discovery of adversarial examples (Goodfellow et al., 2014), constructing adversarially robust models has become one of the most studied research topics (Mao et al., 2023; Ilyas et al., 2019; Schmidt et al., 2018). Among various existing defense strategies, adversarial training has been found to be the most effective approach against adversarial attacks (Mao et al., 2023; Wang & Wang, 2022; Qin et al., 2019; Goodfellow et al., 2014). Additionally, there is another prominent category of defense methodologies known as certified defense. These methods usually provide theoretically guaranteed bounds on the model's adversarial robustness (Zhang et al., 2022a; 2021; Jia et al., 2019; Gowal et al., 2018; Yoshida & Miyato, 2017). For example, Gowal et al. (2018) employs the interval bound propagation to achieve fast and stable verified robust training. On the other hand, despite the great importance of studying adversarial robustness from the unlearning perspective, there is no existing work exploring the adversarial robustness properties of the unlearned models in the context of the right to be forgotten. Although Hu et al. (2023); Qian et al. (2023); Di et al. (2022) study potential pitfalls during unlearning, their emphasis is merely on crafting malicious perturbations to perform data poisoning attacks. To the best of our knowledge, no prior research has examined the adversarial risks associated with the standard unlearning process from the perspective of adversarial attacks.

## 3 ADVERSARIAL UNLEARNING ATTACK

### 3.1 BUILDING MOTIVATION

From Definition 1, we know that machine unlearning provides a solution to mitigate these privacy risks, and involves the design of sophisticated unlearning techniques to remove private information from a trained machine learning model. Despite the focus on safeguarding individuals' private and sensitive data, existing unlearning methods neglect to assess the vulnerability of the unlearned models to adversarial attacks and security breaches. This oversight raises concerns regarding the adversarial robustness and overall security of the unlearning process. As no previous literature has studied the adversarial vulnerabilities of the unlearning system, for the ***threat model***, we start by discussing the adversary's objectives, capabilities, and knowledge for our attack. The objective of the adversary is to make malicious unlearning requests to deliberately undermine the adversarial robustness of the unlearned model. The adversary can generate the unlearning requests during the

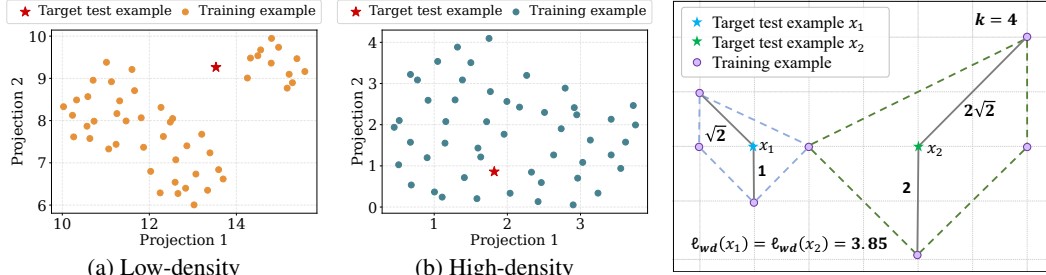

Figure 2: Illustrations of the target test example and training examples in low-density region and high-density region.

Figure 3: An illustration example of distance evaluation metric.

unlearning process. Since the AdvUA does not make any perturbations on the training data, in our main evaluation, we do not require a constraint for a malicious unlearning request as long as it is a training sample. We first consider a white-box setting where the AdvUA adversary knows the original model $W^*$, and subsequently, we delve into the black-box setting.

In the context of the right to be forgotten, *adversarial unlearning robustness* describes the property of an unlearned model to consistently predict the target class label for all perturbed inputs $x'$ in an $l_p$-norm ball $\mathcal{B}_p^{\epsilon_p}(x) = \{x' \in \mathcal{X} : ||x - x'||_p \leq \epsilon_p\}$ of radius $\epsilon_p$, after deleting the requested unlearning samples. To formalize this concept, we provide the following definition of adversarial robustness within the context of the right to be forgotten.

**Definition 2** ($A_{S_u}^{\epsilon_p}(x)$-Adversarial Unlearning Robustness). *Consider a sample $x \in \mathcal{X}$, a scalar $\epsilon_p$, and a distance metric $\mathcal{D}(x, x') = ||x - x'||_p$. We use $F(x; W^u)$ to denote the unlearned model, which is derived by removing the requested unlearning samples $S_u$ from the original model $W^*$. The unlearned model $F(x; W^u)$ is robust to adversarial perturbations of magnitude $\epsilon_p$ at input $x$ if and only if $\forall (x + \delta) \in \mathcal{B}_p^{\epsilon_p}(x)$, $\arg\max_{c \in \mathcal{Y}} F(x; W^u)[c] = \arg\max_{c \in \mathcal{Y}} F(x + \delta; W^u)[c]$, where $W^u = U_L(S, L(S), S_u)$, and $\delta \in \mathbb{R}^d$ is the adversarial perturbation for $x$.*

Our goal in this paper is to investigate whether the process of unlearning hinders an unlearned model's ability to withstand adversarial perturbations. Specifically, in this paper, we motivate our study with the previously raised scientific questions (i.e., *Q1* and *Q2* in Section 1). In Definition 2, we allow a bounded worst-case perturbation before passing the perturbed sample to the unlearned model $W^u$ in the context of the right to be forgotten. In particular, based on the above definition, we can obtain that when $\epsilon_p = 0$ and $\arg\max_{c \in \mathcal{Y}} F(x; W^u)[c] \neq \arg\max_{c \in \mathcal{Y}} F(x + \delta; W^u)[c]$, *the attacker misleads the unlearned model $W^u$ to directly misclassify the target victim samples without any further perturbations*. Without loss of generality, we set $p = \infty$ in this paper and omit this subscript for simplicity of notations. Additionally, using Definition 2, we can easily estimate the adversarial unlearning robustness of the unlearned model $W^u$ across a population of data samples.

### 3.2 FORMULATION OF ADVUA

As previously discussed (see Fig. 1), we have observed that the adversarial robustness of unlearned models can be substantially compromised even with random data removal. To gain a deeper understanding, we now delve into examining how unlearning samples affect both successful and unsuccessful adversarial examples. Then we present two visual examples that demonstrate the successful adversarial example in Fig. 2a and the unsuccessful adversarial example in Fig. 2b after unlearning. Note that the two target test examples in Fig. 2 cannot be adversarially attacked with the same perturbation sets *before unlearning*. Here we utilize the last representation layer and apply UMAP (McInnes et al., 2018) to project the adversarial examples, along with their 50 nearest neighbors. From this figure, we can observe that the target victim example shown in Fig. 2a has now transitioned to a low-density region, and can easily fool the unlearned model. In contrast, the target victim example depicted in Fig. 2b remains situated within a high-density region, making it difficult to generate a successful adversarial example for this particular target example.

**Low-density Regions Generation.** Drawing on the aforementioned observations, we make the following key contribution to exacerbating adversarial vulnerability in the unlearned models: employing unlearning techniques to strategically position target victim samples within low-density regions. As a result, the unlearned models may not have learned robust decision boundaries or patterns for these regions, making it more susceptible to adversarial attacks and misclassifications. Let $\mathcal{V} = \{(x_v, y_v)\}_{v=1}^V$ represent the victim samples that adversaries intend to attack. We use

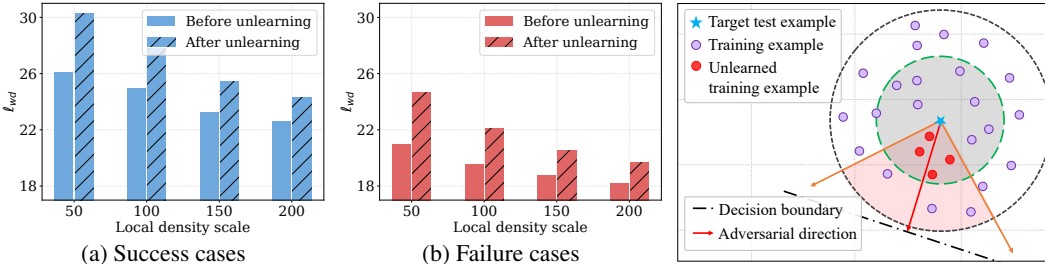

Figure 4: Density of success and failure cases before and after unlearning measured by distance evaluation metric on different local scales. A high $\ell_{wd}$ value indicates a low-density region.

Figure 5: Illustration of adversarial direction alignment for unlearning training examples.

$\mathcal{N}_{K_v}(x_v)$ to denote the top-$K_v$ nearest neighbors for the target sample $x_v$. Let $x_v^{max} \in \mathcal{N}_{K_v}(x_v)$ denote the sample that has the largest distance from the target victim sample $x_v$. To estimate the space-filling capability of the region surrounding sample $x_v$, we compute the relative distance information of its neighbors, with respect to $x_v^{max}$, utilizing classical expansion models (Karger & Ruhl, 2002; Houle et al., 2012; Houle, 2017; Ma et al., 2018). Specifically, for $x_v$, the relative distance of $x_i \in \mathcal{N}_{K_v}(x_v)$ with respect to $x_v^{max}$ in the representation space is calculated as $\log(||H(x_i; W^u) - H(x_v; W^u)||_2/||H(x_v^{max}; W^u) - H(x_v; W^u)||_2)$. However, if we directly calculate this distance value, we may encounter the case where $\forall x_i \in \mathcal{N}_{K_v}(x_v), ||H(x_i; W^u) - H(x_v; W^u)||_2 = ||H(x_v^{max}; W^u) - H(x_v; W^u)||_2$. On the other hand, relying solely on relative distances can be challenging, as they do not provide insights into the absolute values of data points. As illustrated in Fig. 3, even though $x_1$ and $x_2$ have the same relative distance values, $x_2$ is more likely to be vulnerable to adversarial attacks. Therefore, in addition to the relative distances, we should also take absolute distance information into account. For $x_v$ and its neighbors $\mathcal{N}_{K_v}(x_v)$, we can calculate the Euclidean distance between this sample and its neighbor $x_i \in \mathcal{N}_{K_v}(x_v)$ as $D(H(x_i; W^u), H(x_v; W^u)) = ||H(x_i; W^u) - H(x_v; W^u)||_2^2$, where $H(x_i; W^u)$ is the representation of $x_i$. By combining the above, we formulate the following measure

$$\ell_{wd}(x_v, \mathcal{N}_{K_v}(x_v); W^u) = -\lambda_1 \left( \frac{1}{K_v} \sum_{x_i \in \mathcal{N}_{K_v}(x_v)} \log \frac{||H(x_i; W^u) - H(x_v; W^u)||_2}{||H(x_v^{max}; W^u) - H(x_v; W^u)||_2 + \beta} \right)^{-1}$$

$$+ \frac{(1 - \lambda_1)}{K_v} \sum_{x_i \in \mathcal{N}_{K_v}(x_v)} D(H(x_i; W^u), H(x_v; W^u)), \tag{1}$$

where $\lambda_1 \in [0, 1]$ and $K_v = |\mathcal{N}_{K_v}(x_v)|$. In the above, $\beta$ is a pre-defined value to avoid cases where the fraction has a zero denominator. The first term in $\ell_{wd}$ is used to quantify the local density around $x_i$ measured in the space subsequent to unlearning $S_u$ (Houle, 2017; Ma et al., 2018; Zhang et al., 2019). The second one ensures a comprehensive characterization of the local behavior around $x_v$.

To assess the impact of unlearning on density as expressed in Eqn. (1), we focus on *hard samples* that are attacked successfully (yielding incorrect predictions) in standard training but failed (resulting in correct predictions) in adversarial training. In Fig. 4, we compare the average density of success cases (attack success after unlearning) and failure cases (attack fail after unlearning) within different local density scales (nearest neighbors). More details can be found in Appendix E. Fig. 4 shows that success cases typically have higher $\ell_{wd}$ compared to failure cases after unlearning, which suggests that success cases locate in relatively low-density regions and are thus susceptible to attacks (consistent with observations in Fig. 2).

**Attack Direction Alignment.** From the above, we know that for a target sample $x_v$ located in a low-density region, it tends to experience successful attacks during inference since it is less likely to be covered by the distribution of the training samples. The next question we want to explore is whether all of the nearest samples are equally critical for performing adversarial unlearning attacks to assign a wrong label to $x_v$. However, in this work, we find an intriguing phenomenon: not all of the nearest samples are equally important for adversarial unlearning attacks. This phenomenon reveals that for $x_v$, the efficient unlearning directions towards the low-density regions should align well with its adversarial attack direction, which is determined by $\nabla_{x_v} \mathcal{L}(x_v, y_v; W)$, where $y_v$ is the label of $x_v$. To help understanding, we give an illustration example for the attack direction alignment in Fig. 5, where the grey area represents the high-density region for the target test example and red zones highlight directions that are positively aligned with the red arrow. From this figure, the samples

that fall within the intersection of the grey and red areas are the ones we should focus on. Given sample $x_v$ and its neighbor sample $x_i \in \mathcal{N}_{K_v}(x_v)$, we propose the following measure to estimate the alignment between the unlearning attack and the adversarial attack

$$\ell_{dirc}(x_i, x_v; W^u) = \frac{(H(x_i; W^u) - H(x_v; W^u)) \cdot (\nabla_{x_v}\mathcal{L}(x_v, y_v; W^u))}{||H(x_i; W^u) - H(x_v; W^u)||_2 * ||\nabla_{x_v}\mathcal{L}(x_v, y_v; W^u)||_2}, \quad (2)$$

where $H(x_i; W^u) - H(x_v; W^u)$ is the unlearning direction when we unlearn a sample $x_i \in \mathcal{N}_{K_v}(x_v)$. In the above, we use the normalized adversarial attack to remove the influence of the scaling factor when comparing different models. Based on the equation, for the effective unlearning samples, they should closely match the adversarial attack with the unlearning direction.

**Overall Loss.** Based on the above observations, to understand the worst-case attack performance, we design a novel adversarial unlearning attack framework to increase the inherent adversarial vulnerability of the unlearned models. Let $S_e = \{x_t\}_{t=1}^T$ represent a subset of $S$ that is accessible to the adversary. For each $x_t \in S_e$, we define a discrete indication parameter $\xi_t \in \{0, 1\}$ to indicate whether the sample $x_t$ should be completely deleted ($\xi_t = 1$) or not ($\xi_t = 0$). The forget set $S_u$ to be unlearned is denoted as $S_u = S_e \circ \Phi = \{x_t | x_t \in S_e \text{ and } \xi_t = 1\}$, where $\Phi = \{\xi_t\}_{t=1}^T$. We use $\{(x_v, y_v)\}_{v=1}^V$ to denote the *hard* target samples, which cannot be successfully attacked from the original model $W^*$ using the same perturbation budget $\epsilon$. Based on the above, to effectively attack these hard target samples (i.e., $\{(x_v, y_v)\}_{v=1}^V$), we formulate the below overall loss

$$\max_{\{\delta_v \in \mathcal{B}^\epsilon(x_v)\}_{v=1}^V} \sum_{v=1}^V \mathcal{L}(x_v + \delta_v, y_v; W^u(\Phi)) \quad (3)$$

$$s.t. \Phi \leftarrow \arg\max_{\Phi} \sum_{v=1}^V \ell_{wd}(x_v, \mathcal{N}_{K_v}(x_v); W^u(\Phi)) - \sum_{v=1}^V \frac{\lambda_2}{K_v} \sum_{x_i \in \mathcal{N}_{K_v}(x_v)} \ell_{dirc}(x_i, x_v; W^u(\Phi)),$$

where $K_v = |\mathcal{N}_{K_v}(x_v)|$ and $W^u(\Phi) = U_L(S, W^*, S_u = S_e \circ \Phi)$. $\mathcal{L}$ is the loss function to enforce the adversarial example $x_v + \delta_v$ to be predicted as a different label than $y_v$. Exactly solving Eqn. (3) would be computationally infeasible (Korte et al., 2011), and we instead refer to an empirical greedy to solve the above optimization problem (Barron et al., 2008). Algorithm 1 in Appendix A details the procedure to optimize the above formulated overall loss. *Notably, in the above, when $\delta_v = 0$, the requested malicious unlearning samples can cause the unlearned model $W^u$ to directly misclassify sample $x_v$ without the need for any further adversarial manipulations.*

**Theorem 1.** *Consider a data distribution $\mathcal{X}$ characterized by a Gaussian distribution with mean $\mu \in \mathbb{R}^d$ and variance $\sigma^2 I$, i.e., $\mathcal{X} \sim N(\mu, \sigma^2 I)$. Let $\{x_i\}_{i=1}^n$ be a set of samples drawn from $N(\mu, \sigma^2 I)$. Then the expected local density around point $\tilde{x}$ is lower bounded by*

$$\mathbb{E}_{\{x_i\}_{i=1}^n \sim \mathcal{X}} \left[ \sum_{i=1}^n \mathbf{1}\left\{ ||x_i - \tilde{x}||_2^2 \leq q \right\} \right] \geq n \times \left[ 1 - \frac{\sigma^2 d}{(q - ||\mu - \tilde{x}||_2^2)^2} \right] \quad (4)$$

*where $\tilde{x} \in \mathbb{R}^d$ and $q \in \mathbb{R}$.*

**Theorem 2.** *Let $g_n$ be any learning algorithm, i.e., a function from $n \geq 0$ samples in $\mathbb{R}^d \times \{\pm 1\}$ to a binary classifier $f_n$. Moreover, let $W \in \mathbb{R}^d$ be the weight of $f_n$ and $W = \frac{1}{n}\sum_{i=1}^n y_i x_i$, and let $\theta \in \mathbb{R}^d$ be drawn from $N(0, I)$, $||\theta||_2 = \sqrt{d}$. We draw $n_1, n_2$ samples from the $(\theta, \sigma)$-Gaussian model, which generates $(x, y) \in \mathbb{R}^d \times \{\pm 1\}$ by first randomly selecting a label $y \in \{\pm 1\}$ and then sampling $x \in \mathbb{R}^d$ from $N(y \cdot \theta^\star, \sigma^2 I)$. Let the expected $l_\infty^\epsilon$-robust classification errors of $f_{n_1}, f_{n_2}$ are $R_1, R_2$. Then it can be deduced that $R_1 \leq R_2$ holds with a probability at least $1 - 2\exp\left(-\frac{d}{8(\sigma^2+1)}\right)$ if $n_1 \geq c_2\epsilon^2\sqrt{d}$ and $n_2 \leq \frac{\epsilon^2\sigma^2}{8\log d}$, where $0 \leq \sigma \leq c_1 d^{1/4}$, $\sqrt{\frac{8\log d}{\sigma^2}} \leq \epsilon \leq \frac{1}{2}$.*

In Theorem 1, we estimate the local density at a given point $\tilde{x}$ by counting the number of data covered within a ball centered as $\tilde{x}$ with radius $q$. This theorem demonstrates that as the quantity of training data decreases, the average local density also experiences a proportional decline. Theorem 2 illustrates the relation between the number of training samples and the associated $l_\infty^\epsilon$-adversarial robust error. This theorem shows that when the training set size diminishes from $n_1$ to $n_2$ under the conditions mentioned in this theorem, there is a high probability that the robust error will increase. *The proofs of Theorem 1 and Theorem 2 are deferred to Appendix B.*

**Discussions on the Black-box Setting.** The black-box attack assumes that the adversary only knows the output of the target model through predictions. The black-box attack can be executed by constructing a surrogate model and transferring adversarial examples, which leverages shared decision

boundaries among various models. This approach can reduce exposure risk, and hence, many existing works (Byun et al., 2022; Zhou et al., 2018; Liu et al., 2017) have been focused on adversarial transferability. In our black-box setting, we can train several surrogate models and transfer both the selected unlearning samples and generated adversarial examples to attack the target victim model.

**Enhanced Model Stealing Attacks.** Traditional model stealing attacks (Genç et al., 2023; Yu et al., 2020; Tramèr et al., 2016) first send a series of queries $Q$ to the target victim model and then use the collected query-response pairs to train a surrogate model that approximates the behavior of the victim model. The key to the success of model stealing is to find query samples lying approximately on the decision boundary of the victim model, which is not easy. To improve the query effectiveness, our goal here is to make malicious unlearning requests to learn the decision boundary information of the victim target model $W^*$. Instead of starting from scratch, following existing works, we first construct an initialized primitive substitute model. Here, we consider the scenarios where the attacker has a set of query samples $Q$ for querying the victim target model. In the context of machine unlearning, we also assume the attacker has available unlearning samples, denoted as $S_e$. Based on AdvUA (in Eqn. (3)), we can select a set of malicious unlearning samples to increase its query effectiveness by revealing the important decision boundary information of the victim model.

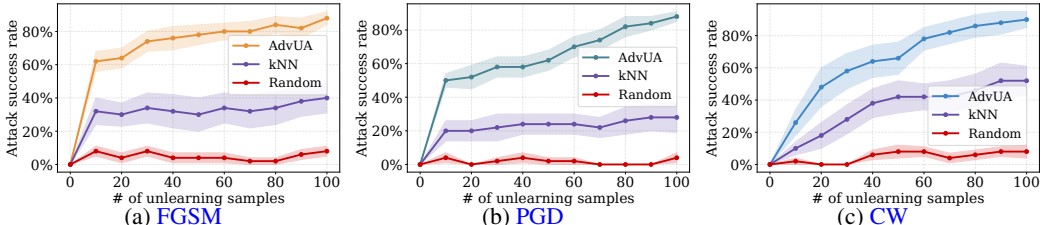

(a) FGSM  (b) PGD  (c) CW

Figure 6: Attack performance of AdvUA on adversarially trained models against various attacks.

# 4 EXPERIMENTAL RESULTS

In this section, we conduct experiments to validate the performance of AdvUA. All experiments are performed for 10 independent trials, and we report the mean and standard errors in the following analyses. *For more experimental details (e.g., experimental setup and parameter settings) and experimental results (e.g., more unlearning methods and ablation studies), please refer to Appendix E.*

## 4.1 EXPERIMENTAL SETUP

**Datasets and Models.** In experiments, we adopt the following datasets: ImageNet (Deng et al., 2009), CIFAR-10 (Krizhevsky & Hinton, 2010), SVHN (Netzer et al., 2011), and IRIS (Fisher, 1988). We consider various deep learning models, including ResNet-50, ResNet-18 (He et al., 2016), DenseNet-121 (Huang et al., 2017), VGG-19 (Simonyan & Zisserman, 2015), a 5-layer ConvNet with max-pooling and dropout, and a multilayer perceptron (MLP).

**Baselines.** We compare the performance of AdvUA with the following baselines: *random* deletion and *k-nearest neighbors (kNN)* deletion. The random deletion method randomly removes some training samples from the training set, regardless of the target sample. The kNN deletion method removes the $k$ closest training samples to a target sample in the input space.

**Implementation Details.** We conduct comprehensive experiments to evaluate the attack performance of AdvUA on both undefended and defended models. The undefended models are constructed using natural training algorithms, and the defended models are constructed using adversarial training and certified defense methods. The adversarial training involves training the model against a PGD adversary with $l_\infty$ project gradient descent of $\epsilon = 8/255$ (Madry et al., 2018). The certified defense methods utilize techniques including spectral norm regularization (Yoshida & Miyato, 2017) and interval bound propagation (Gowal et al., 2018) to provide provably robustness bounds. We evaluate the robustness of undefended and defended models against FGSM (Goodfellow et al., 2015), PGD (Madry et al., 2018), and CW (Carlini & Wagner, 2017) adversarial attacks. Specifically, we use a perturbation budget of $\epsilon = 8/255$ to generate adversarial examples for each attack, and we set steps to 7 for the PGD attack and steps to 30 for the CW attack. Regarding the unlearning methods for removing the training samples, we select the first-order based unlearning method (Warnecke et al., 2023) and SISA (Bourtoule et al., 2021a).

## 4.2 ATTACK PERFORMANCE AGAINST ROBUST TRAINING

**Attack Performance against Adversarial Training.** We first investigate the attack performance of AdvUA on the defended models against FGSM, PGD, and CW attacks. The defended models are

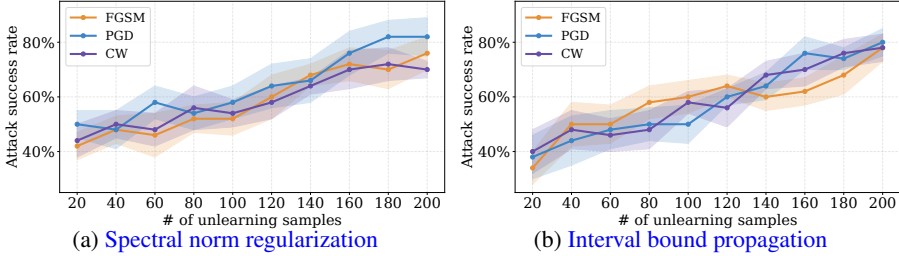

Figure 7: Attack performance of AdvUA on certified defended models against various attacks.

adversarially trained with ResNet-18 on CIFAR-10 and ConvNet on SVHN. The results are shown in Figs. 6a and 6b for CIFAR-10, and Fig. 6c for SVHN. Here, we randomly select hard samples that are correctly classified on the defended model and then remove various quantities of training samples for each sample using the first-order based unlearning method. As shown in the figures, AdvUA significantly enhances the attack success rates on the defended models. The increase in vulnerability can be primarily because, after the unlearning process, test samples shift to low-density regions that adversarially trained models struggle to cover effectively, especially along the adversarial direction. As more training samples are removed, these test samples migrate to even sparser regions, resulting in higher attack success rates. For instance, when unlearning 100 training samples for each hard sample, AdvUA achieves approximately $88\%$ attack success rates on CIFAR-10 against FGSM and PGD attacks and $90\%$ on SVHN against the CW attack. In comparison, the attack performance of the random baseline is notably poor. While the kNN baseline slightly improves, it still significantly lags behind AdvUA. One reason is that although the kNN might reduce local density to a degree, it is not as effective as AdvUA. Additionally, AdvUA uniquely aligns the direction of unlearning with the adversarial direction. Therefore, this experiment illustrates the efficacy and superiority of AdvUA in selecting crucial unlearning samples to achieve the desired attack goals.

**Attack Performance against Certified Defense.** Next, we explore the attack performance of AdvUA on the certified defended models using spectral norm regularization and interval bound propagation techniques. In spectral norm regularization, we adopt the VGG model in Yoshida & Miyato (2017) and choose $\epsilon = 2/255$. In interval bound propagation, we adopt the small CIFAR-10 model in Gowal et al. (2018) and choose $\epsilon = 8/255$. Fig. 7 shows the experimental results of AdvUA on the certified defended models against FGSM, PGD, and CW attacks on CIFAR-10. As we can observe, AdvUA performs reasonably well against these certified methods. Notably, achieving a high attack success rate necessitates unlearning more training samples than needed in adversarial training. One reason could be that certified defense methods are generally less sensitive to the specific training samples used, in contrast to the more data-sensitive nature of adversarial training.

## 4.3 ATTACK TRANSFERABILITY AND MODEL STEALING ATTACKS

**Effectiveness on Attack Transferability.** We consider a black-box scenario to examine the effectiveness of AdvUA on adversarial transferability. We use a surrogate model to unlearn various numbers of training samples with the first-order based method and then generate the adversarial examples. Sub-

Table 1: Attack transferability of FGSM with AdvUA.

| Method | # of unlearning samples | VGG-19 | DenseNet-121 |
|---|---|---|---|
| Baseline | None | $63.0\% \pm 4.7\%$ | $61.0\% \pm 6.9\%$ |
| **AdvUA** | 10 | $\mathbf{88.0\% \pm 2.9\%}$ | $\mathbf{88.0\% \pm 2.9\%}$ |
| | 20 | $\mathbf{88.0\% \pm 2.9\%}$ | $\mathbf{90.0\% \pm 1.5\%}$ |
| | 30 | $\mathbf{92.0\% \pm 2.0\%}$ | $\mathbf{96.0\% \pm 1.6\%}$ |

sequently, these unlearning samples and adversarial examples are transferred to attack the target model. Table 1 presents the attack transferability of FGSM on the undefended models on ImageNet, where we adopt the ResNet-50 as the surrogate model and the VGG-19 and DenseNet-121 as the target models. For comparison, we also include the attack transferability of the baseline method, which directly applies generated adversarial samples to attack the target models (without unlearning). Conversely, AdvUA employs unlearning to create a sparse local density environment around the test sample prior to the transfer process. The results show that AdvUA outperforms the baseline by a large margin in both target models. For instance, when transferring ResNet-50 to DenseNet-121, AdvUA achieves a $96\%$ attack success rate with 30 unlearning samples, while the baseline only receives a $61\%$ attack success rate. These findings clearly indicate that AdvUA effectively boosts attack transferability in the realm of adversarial machine learning.

**Effectiveness on Model Stealing Attacks.** We also evaluate the effectiveness of AdvUA on model stealing attacks. We initially train a target model composed of an MLP using the IRIS dataset and then employ a synthetic dataset as queries to steal this model, as outlined in

Table 2: Test accuracy of model stealing attacks.

| Query count | Query-based attack | Query-based attack + unlearn (AdvUA) |
|---|---|---|
| 100 | $45.24\% \pm 6.17\%$ | $\mathbf{85.71\% \pm 4.92\%}$ $(\mathbf{1.89\times})$ |
| 200 | $56.19\% \pm 9.85\%$ | $\mathbf{88.57\% \pm 2.41\%}$ $(\mathbf{1.58\times})$ |
| 300 | $85.56\% \pm 2.81\%$ | $\mathbf{90.00\% \pm 2.85\%}$ $(\mathbf{1.05\times})$ |
| 400 | $86.11\% \pm 5.74\%$ | $\mathbf{92.22\% \pm 1.41\%}$ $(\mathbf{1.07\times})$ |

Tramèr et al. (2016). Table 2 shows the test accuracy of the extracted model obtained with and without unlearning. The numbers in the parentheses represent the improvement of test accuracy by integrating AdvUA. The results indicate that our method boosts the performance of model stealing attacks, especially when queries are limited. For instance, AdvUA enhances the test accuracy of the extracted model by a factor of $1.89\times$ with 100 queries compared to the original model stealing attack. By strategically unlearning samples to diminish the local density near the target query, especially close to the decision boundary, we create a localized region with reduced robustness around the target query. This adjustment makes the query more effective in training the derived model, leading to superior accuracy compared to the original query-based attack. In summary, AdvUA effectively improves the performance of model stealing attacks, providing another perspective to validate the influence of unlearning on model robustness.

## 4.4 ABLATION STUDY

In this section, we conduct ablation studies over sampling density, perturbation bound, and unlearning methods. As depicted in Fig. 8a, when unlearning the same number of training samples, test samples initially from low-density regions exhibit higher PGD attack success rates than those from high-density regions. Notably, even when selecting test samples from high-density regions, AdvUA still achieves remarkable outcomes. We also compare attack success rates with different perturbation bounds during CW attack on CIFAR-10. As shown in Fig. 8b, despite larger perturbation bounds benefiting baseline methods, AdvUA consistently outperforms them. In addition, Fig. 8c illustrates the effectiveness of AdvUA with the SISA unlearning method, to attack the defended model against FGSM, PGD, and CW attacks on CIFAR-10. For example, AdvUA can achieve around $65\%$ PGD attack success rates with 50 unlearning samples. In these ablation studies, our experimental results emphasize the efficacy of AdvUA in increasing the defended model's vulnerability, irrespective of variations in sampling density, perturbation bounds, and unlearning techniques.

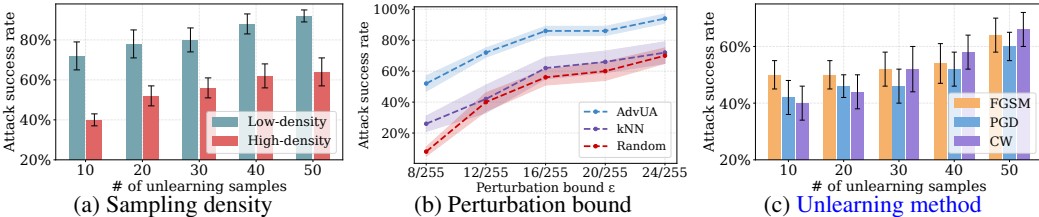

(a) Sampling density      (b) Perturbation bound      (c) Unlearning method

Figure 8: Ablation studies over sampling density, perturbation bound, and unlearning method.

## 5 CONCLUSION AND FUTURE WORK

In this paper, we take a significant step towards developing a comprehensive understanding of the adversarial risks associated with the process of machine unlearning. We show that the low density of the target sample in the space, along with the alignment between adversarial attacks and unlearning directions, are crucial factors for generating successful adversarial examples, which were not achievable prior to unlearning. Drawing upon our insightful observations, in this paper, we design a new adversarial unlearning attack (AdvUA), with the ultimate goal of exacerbating the adversarial vulnerability of the unlearned models. What's more, AdvUA can also make model stealing attacks more effective and stealthy (i.e., requiring fewer query samples). Our extensive experimental results serve as strong empirical evidence of the effectiveness of the proposed methodology, underscoring the importance of considering security implications alongside data privacy concerns within the domain of machine unlearning.

In the future, we will investigate the detection and defense mechanisms to mitigate and defend against adversarial unlearning attacks in the context of the right to be forgotten. Besides deep learning models, we will also investigate the potential threats of adversarial unlearning attacks in other domains (e.g., federated learning, and graph neural networks) using different unlearning methods.

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

# APPENDIX

## A   ALGORITHM

In Algorithm 1, we detail the procedure to find the optimal forget set for the adversarial unlearning attacks. We formulate the optimization problem in the main manuscript as an empirical search task. Given a set of available training samples $S_e$, we first split $S_e$ into $Z$ subsets, i.e., $\{S_e^z\}_{z \in \mathcal{Z}}$, where $\mathcal{Z} = [Z]$. In each iteration, we aim to find a training subset $S_e^z$, when removed, that contributes most to reducing the local density and matching the adversarial attack direction of the target samples. Specifically, in each iteration, we unlearn each training subset $S_e^z$ and compute the loss associated with the distance evaluation metric and cosine similarity. Then, we add the training samples from the subset with the maximum loss to the forget set $S_u$, remove the subset from subsets of $S_e$, and update the well-trained model $W^*$ by unlearning these particular training samples. After this, we can proceed to the next iteration (i.e., find the next effective subset) until we reach the maximum forget budget.

---

**Algorithm 1** Constructing Adversarial Unlearning Attack Algorithm

---

**Input:** Well-trained model $F(W^*)$, training data $S$, available training samples $S_e = \{x_t\}_{t=1}^T$, unlearning algorithm $U_L$, forget set budget $B \leq T$, number of subsets $Z$, hard target samples $\{(x_v, y_v)\}_{v=1}^V$

**Output:** Forget set $S_u$
  1: Initialize $S_u \leftarrow \emptyset$
  2: Split $S_e$ into subsets $\{S_e^z\}_{z \in \mathcal{Z}}$, where $\mathcal{Z} = \{1, \cdots, Z\}$
  3: **while** $|S_u| \leq B$ **do**
  4:   **for each** $z \in \mathcal{Z}$ **do**
  5:     $\tilde{W} = W^*$
  6:     $W^u = U_L(S, \tilde{W}, S_e^z)$
  7:     Get top-$K_v$ nearest neighbors $\mathcal{N}_{K_v}(x_v)$
  8:     $\ell^z = \sum_{v=1}^V \ell_{wd}(x_v, \mathcal{N}_{K_v}(x_v); W^u) - \sum_{v=1}^V \frac{\lambda_2}{K_v} \sum_{x_i \in \mathcal{N}_{K_v}(x_v)} \ell_{dirc}(x_i, x_v; W^u)$
  9:   **end for**
 10:   $\tilde{z} \leftarrow \arg\max_{z \in \mathcal{Z}} \ell^z$
 11:   $S_u \leftarrow S_u \cup S_e^{\tilde{z}}$
 12:   $\mathcal{Z} \leftarrow \mathcal{Z} \setminus \tilde{z}$
 13:   $W^* \leftarrow U_L(S, W^*, S_e^{\tilde{z}})$
 14: **end while**

---

## B   PROOFS

### B.1   PROOF OF THEOREM 1

**Theorem 1.** *Consider a data distribution $\mathcal{X}$ characterized by a Gaussian distribution with mean $\mu \in \mathbb{R}^d$ and variance $\sigma^2 I$, i.e., $\mathcal{X} \sim N(\mu, \sigma^2 I)$. Let $\{x_i\}_{i=1}^n$ be a set of samples drawn from $N(\mu, \sigma^2 I)$. Then the expected local density around point $\tilde{x}$ is lower bounded by*

$$\mathbb{E}_{\{x_i\}_{i=1}^n \sim \mathcal{X}} \left[ \sum_{i=1}^n \mathbf{1}\left\{ \|x_i - \tilde{x}\|_2^2 \leq q \right\} \right] \geq n \times \left[ 1 - \frac{\sigma^2 d}{(q - \|\mu - \tilde{x}\|_2^2)^2} \right] \quad (5)$$

*where $\tilde{x} \in \mathbb{R}^d$ and $q \in \mathbb{R}$.*

*Proof.* To begin with, the expectation $\mathbb{E}_{\{x_i\}_{i=1}^n \sim \mathcal{X}} \left[ \sum_{i=1}^n \mathbf{1} \left\{ \|x_i - \tilde{x}\|_2^2 \leq k\sigma \right\} \right]$ is expressed as the expected value of a binomial distribution with $N$ trials. Then we can have

$$
\begin{aligned}
&\mathbb{E}_{\{x_i\}_{i=1}^n \sim \mathcal{X}} \left[ \sum_{i=1}^n \mathbf{1} \left\{ \|x_i - \tilde{x}\|_2^2 \leq k\sigma \right\} \right] \\
&= \sum_{i=1}^n \mathbb{E}_{x_i \sim \mathcal{X}} \left[ \mathbf{1} \left\{ \|x_i - \tilde{x}\|_2^2 \leq k\sigma \right\} \right] \\
&= n \times P \left( \|x_i - \tilde{x}\|_2^2 \leq k\sigma \right) \\
&= n \times P \left( \|x_i - \mu + \mu - \tilde{x}\|_2^2 \leq k\sigma \right) \\
&\geq n \times P \left( \|x_i - \mu\|_2^2 + \|\mu - \tilde{x}\|_2^2 \leq k\sigma \right) \\
&= n \times P \left( \|x_i - \mu\|_2^2 \leq k\sigma - \|\mu - \tilde{x}\|_2^2 \right) \\
&= n \times P \left( \|x_i - \mu\|_2^2 \leq \sigma \cdot \left( k - \frac{1}{\sigma} \|\mu - \tilde{x}\|_2^2 \right) \right) \\
&= n \times P \left( \sqrt{(x_i - \mu)^\top \sigma^{-2} I (x_i - \mu_i)} \leq k - \frac{1}{\sigma} \|\mu - \tilde{x}\|_2^2 \right).
\end{aligned}
\tag{6}
$$

Upon applying the Multidimensional Chebyshev's Inequality (Chen, 2007), we can obtain

$$
\mathbb{E}_{\{x_i\}_{i=1}^n \sim \mathcal{X}} \left[ \sum_{i=1}^n \mathbf{1} \left\{ \|x_i - \tilde{x}\|_2^2 \leq k\sigma \right\} \right] \geq n \times \left[ 1 - \frac{d}{\left( k - \frac{1}{\sigma} \|\mu - \tilde{x}\|_2^2 \right)^2} \right].
\tag{7}
$$

Let $q = k\sigma$, then we can get

$$
\mathbb{E}_{\{x_i\}_{i=1}^n \sim \mathcal{X}} \left[ \sum_{i=1}^n \mathbf{1} \left\{ \|x_i - \tilde{x}\|_2^2 \leq q \right\} \right] \geq n \times \left[ 1 - \frac{\sigma^2 d}{\left( q - \|\mu - \tilde{x}\|_2^2 \right)^2} \right].
\tag{8}
$$

$\square$

## B.2 PROOF OF THEOREM 2

**Lemma 1.** *Let $(x_1, y_1), \ldots, (x_n, y_n)$ be drawn i.i.d. from a $(\theta^\star, \sigma)$-Gaussian model with $\|\theta^\star\|_2 = \sqrt{d}$ and $\sigma \leq \frac{1}{32} d^{1/4}$. Let $W \in \mathbb{R}^d$ be the weighted mean vector $W = \frac{1}{n} \sum_{i=1}^n y_i x_i$. Then with probability at least $1 - 2\exp\left( -\frac{d}{8(\sigma^2+1)} \right)$, the linear classifier $f_W$ has $l_\infty^\epsilon$-robust classification error at most $\frac{1}{2}(1 - 1/d)$ if*

$$
n \geq \begin{cases} 1 & \text{for } \epsilon \leq \frac{1}{3} d^{-1/4} \\ 16\epsilon^2 \sqrt{d} & \text{for } \frac{1}{3} d^{-1/4} \leq \epsilon \leq \frac{1}{2} \end{cases}.
\tag{9}
$$

*Proof.* We begin by invoking Theorem 21 in Schmidt et al. (2018), which gives a $l_\infty^{\epsilon'}$-robust classification error at most $\beta = \frac{1}{2}(1 - 1/d)$ for

$$
\begin{aligned}
\epsilon' &= \frac{2\sqrt{n} - 1}{2\sqrt{n} + 4\sigma} - \frac{\sigma \sqrt{2 \log 1/\beta}}{\sqrt{d}} \\
&\geq \frac{2\sqrt{n} - 1}{2\sqrt{n} + \frac{1}{8} d^{1/4}} - \frac{\sqrt{2 \log \frac{2d}{d-1}}}{32 d^{1/4}} \\
&\geq \frac{2\sqrt{n} - 1}{2\sqrt{n} + \frac{1}{8} d^{1/4}} - \frac{1}{16 d^{1/4}}.
\end{aligned}
\tag{10}
$$

First, we consider the case where $\epsilon \leq \frac{1}{3}d^{-1/4}$. Using $n = 1$, the resulting robustness is

$$
\begin{aligned}
\epsilon' &\geq \frac{1}{2 + \frac{1}{8}d^{1/4}} - \frac{1}{16d^{1/4}} \\
&\geq \frac{1}{\left(2 + \frac{1}{8}\right)d^{1/4}} - \frac{1}{16d^{1/4}} \\
&\geq \frac{111}{272}d^{-1/4} \geq \frac{1}{3}d^{-1/4} \geq \epsilon
\end{aligned}
\tag{11}
$$

as required. Next, we consider the case $\frac{1}{3}d^{-1/4} \leq \epsilon \leq \frac{1}{2}$. Substituting $n = 16\epsilon^2\sqrt{d}$, we get

$$
\begin{aligned}
\epsilon' &\geq \frac{8\epsilon d^{1/4} - 1}{8\epsilon d^{1/4} + \frac{1}{8}d^{1/4}} - \frac{1}{16d^{1/4}} \\
&\geq \frac{5\epsilon d^{1/4}}{4d^{1/4} + \frac{1}{8}d^{1/4}} - \frac{1}{16d^{1/4}} \\
&\geq \frac{40}{33}\epsilon - \frac{3}{16}\epsilon \geq \epsilon.
\end{aligned}
\tag{12}
$$

$\square$

**Theorem 2.** *Let $g_n$ be any learning algorithm, i.e., a function from $n \geq 0$ samples in $\mathbb{R}^d \times \{\pm 1\}$ to a binary classifier $f_n$. Moreover, let $W \in \mathbb{R}^d$ be the weight of $f_n$ and $W = \frac{1}{n}\sum_{i=1}^n y_i x_i$, and let $\theta \in \mathbb{R}^d$ be drawn from $N(0, I)$, $\|\theta\|_2 = \sqrt{d}$. We draw $n_1, n_2$ samples from the $(\theta, \sigma)$-Gaussian model, which generates $(x, y) \in \mathbb{R}^d \times \{\pm 1\}$ by first randomly selecting a label $y \in \{\pm 1\}$ and then sampling $x \in \mathbb{R}^d$ from $N(y \cdot \theta^\star, \sigma^2 I)$. Let the expected $l_\infty^\epsilon$-robust classification errors of $f_{n_1}, f_{n_2}$ are $R_1, R_2$. Then it can be deduced that $R_1 \leq R_2$ holds with a probability at least $1 - 2\exp\left(-\frac{d}{8(\sigma^2+1)}\right)$ if $n_1 \geq 16\epsilon^2\sqrt{d}$ and $n_2 \leq \frac{\epsilon^2\sigma^2}{8\log d}$, where $0 \leq \sigma \leq \frac{1}{32}d^{1/4}$, $\sqrt{\frac{8\log d}{\sigma^2}} \leq \epsilon \leq \frac{1}{2}$.*

*Proof.* We begin by invoking Corollary 23 in Schmidt et al. (2018), which provides an upper bound on the error, denoted as $R_2$, as follows:

$$
R_2 \geq \frac{1}{2}(1 - 1/d) \quad \text{if} \quad n_2 \leq \frac{\epsilon^2\sigma^2}{8\log d}.
\tag{13}
$$

Next we invoke Lemma 1 and analyze the case where $\epsilon \leq \frac{1}{3}d^{-1/4}$. We get

$$
\begin{aligned}
n_2 &\leq \frac{\epsilon^2\sigma^2}{72\log d} \leq \frac{\sigma^2\sqrt{d}}{72\log d} \leq \frac{c_1^2}{72\log d} \\
&\leq \frac{1}{73728\log d} < 1
\end{aligned}
\tag{14}
$$

which is invalid. Therefore, to ensure $n_2 \geq 1$, we need to consider the case where $\frac{1}{3}d^{-1/4} \leq \epsilon \leq \frac{1}{2}$ instead. We have

$$
\begin{aligned}
\frac{\epsilon^2\sigma^2}{8\log d} &\geq 1 \\
\epsilon &\geq \sqrt{\frac{8\log d}{\sigma^2}}
\end{aligned}
\tag{15}
$$

and because $0 \leq \sigma \leq c_1 d^{1/4}$, we also have

$$
\begin{aligned}
\sqrt{\frac{8\log d}{\sigma^2}} &\geq \sqrt{\frac{8\log d}{c_1^2\sqrt{d}}} \\
&\geq 1024d^{-\frac{1}{4}}\sqrt{8\log d} \geq \frac{1}{3}d^{-\frac{1}{4}}.
\end{aligned}
\tag{16}
$$

Therefore, with probability at least $1 - 2\exp\left(-\frac{d}{8(\sigma^2+1)}\right)$, we have $R_1 \leq \frac{1}{2}(1 - 1/d)$ if $n_1 \geq 16\epsilon^2\sqrt{d}$ for $\sqrt{\frac{8\log d}{\sigma^2}} \geq \epsilon \geq \frac{1}{2}$. Hence we can obtain the desired bound that ensures $R_1 \leq \frac{1}{2}(1 - 1/d) \leq R_2$ under the mentioned conditions above. $\square$

**Theorem 3.** *Let* $(x_1, y_1), \ldots, (x_n, y_n) \in \mathbb{R}^d \times \{\pm 1\}$ *be drawn i.i.d. from a* $(\theta^\star, \sigma)$-*Gaussian model with* $\|\theta^\star\|_2 = \sqrt{d}$. *Let* $W \in \mathbb{R}^d$ *be the unit vector in the direction of* $\bar{z} = \sum_{i=1}^n a_i z_i$, $\sum_{i=1}^n a_i = 1$, *i.e.,* $W = \bar{z}/\|\bar{z}\|_2$. *Then with probability at least* $1 - 2\exp\left(-\frac{d}{8(\sigma^2+1)}\right)$, *the linear classifier* $f_W$ *has classification error at most*

$$\exp\left(-\frac{(2\sqrt{n} - \sqrt{\sum_{i=1}^n a_i^2})^2 d}{2(2\sqrt{n}(1+\sigma) + 2\sigma)^2 \sigma^2}\right). \tag{17}$$

*Proof.* Let $z_i = y_i \cdot x_i$ and note that each $z_i$ is independent and has distribution $N(\theta^\star, \sigma^2 I)$. We can derive a conclusion similar to Lemma 16 in Schmidt et al. (2018) as follows:

$$\langle \widehat{w}, \theta^\star \rangle \geq \frac{2\sqrt{n} - \sqrt{\sum_{i=1}^n a_i^2}}{2\sqrt{n}(1+\sigma) + 2\sigma} \sqrt{d} \tag{18}$$

with probability at least $1 - 2\exp\left(-\frac{d}{8(\sigma^2+1)}\right)$. Next, unwrapping the definition of $f_W$ allows us to write the classification error of $f_W$ as

$$\mathbb{P}\left[f_W(x) \neq y\right] = \mathbb{P}\left[\langle W, z_i \rangle \leq 0\right]. \tag{19}$$

We can derive a conclusion similar to Lemma 17 in Schmidt et al. (2018) with $\rho = 0$ then completes the proof of this theorem.

$\square$

By invoking Theorem 3, it is established that with a high probability the derivative of the upper bound of classification error with respect to the quantity of training data remains non-positive. Hence, we can conclude that the upper bound of classification error increases with significant confidence if the sample size decreases.

## C  THE ADOPTED ADVERSARIAL ROBUSTNESS TECHNIQUES

**Adversarial Training.** Adversarial training is recognized as one of the most effective techniques for training robust models to defend against adversarial examples (Cheng et al., 2020; Gao et al., 2019; Madry et al., 2018; Goodfellow et al., 2014). Mathematically, adversarial training can be formulated as a min-max optimization problem, with the goal of finding the optimal solution that remains effective even under the worst-case scenarios. Given a learning model with parameters $W$, a training dataset $S = \{(x_i, y_i)\}_{i=1}^n$, a loss function $\mathcal{L}$, and a perturbation bound $\epsilon$. The optimization problem for training adversarially robust models can be cast as follows

$$\min_W \sum_{i=1}^n \left[\max_{||\delta_i||_p \leq \epsilon} \mathcal{L}(x_i + \delta_i, y_i; W)\right], \tag{20}$$

where $\epsilon$ is pre-defined and $||\cdot||_p$ is the $l_p$ norm distance metric. In this formulation, the inner maximization problem aims to identify the worst-case examples that yield a high loss, while the objective of the outer minimization problem is to adjust model parameters so that the loss in the inner attack is minimized. Madry et al. (2018) propose using projected gradient descent (PGD) to solve the internal maximization problem, the so-called PGD adversarial training, which leverages adversarial samples generated by PGD attacks to train robust models. The PGD attack is a multi-step variant adversary and entails performing projected gradient descent on the negative loss function. Specifically, the adversarial example $x^{t+1}$ at the $(t+1)$-th step for input $x$ is computed as $x^{t+1} = \Pi_{x+\Delta}(x^t + \alpha \cdot sgn(\nabla_x \mathcal{L}(x, y; W)))$, where $\Delta = \{\delta : ||\delta||_p \leq \epsilon\}$ and $\Pi_{x+\Delta}$ is the projection onto the $l_p$ norm-bounded ball of radius $\alpha$ centered at $x$. This dynamic process iteratively generates adversarial examples that challenge the model during training, ultimately leading to increased robustness against adversarial attacks.

**Interval Bound Propagation.** Interval bound propagation is a bounding technique used to train models with proven robustness against adversarial examples (Mirman et al., 2018; Gowal et al.,

2018). The purpose of interval bound propagation is to define a loss that minimizes an upper bound on the maximum difference between any pair of logits when the input can be perturbed within $l_\infty$ norm-bounded ball. More precisely, let $z(x)$ denote the output of a hidden network layer for given an input $x$. We define $\bar{z}(x, \epsilon)$, $\underline{z}(x, \epsilon)$ as interval bounds for $z$ at $x$ with respect to the perturbation bound $\epsilon$ if the following holds for all coordinates $i$

$$\underline{z}(x,\epsilon)[i] \leq \min_{||\delta||_\infty \leq \epsilon} z(x+\delta)[i] \leq \max_{||\delta||_\infty \leq \epsilon} z(x+\delta)[i] \leq \bar{z}(x,\epsilon)[i]. \tag{21}$$

Let $K$ denote the layers in the neural network model, the bounds $\bar{z}_k$ and $\underline{z}_k$ for the $k$-th layer are obtained inductively using simple interval arithmetic. Subsequently, these interval bounds are propagated through all layers of the network. To obtain provable robustness in the context of the classification problem, Gowal et al. (2018) propose the following formulation for the training loss

$$Loss = \lambda_3 \mathcal{L}(z_K, y_{true}) + (1 - \lambda_3)\mathcal{L}(\hat{z}_K(\epsilon), y_{true}), \tag{22}$$

where $\mathcal{L}$ is the cross-entropy loss and $\lambda_3$ is a trade-off parameter. To consider the worst-case prediction and ensure that no perturbation changes the correct class label, the logit of the true class is equal to its lower bound, and the other logits are equal to their upper bound. Therefore, in the above formulation, $\hat{z}_K(\epsilon)[y] = \bar{z}_K(\epsilon)[y]$ if $y \neq y_{true}$, otherwise $\hat{z}_K(\epsilon)[y] = \underline{z}_K(\epsilon)[y_{true}]$.

**Spectral Norm Regularization.** Spectral norm regularization is a regularization technique for reducing the model's sensitivity to small perturbations by penalizing the high spectral norm of weight matrices in neural networks (Cisse et al., 2017; Yoshida & Miyato, 2017; Szegedy et al., 2014). The spectral norm of a matrix measures the largest singular value, which can be thought of as the scaling factor by which the matrix can stretch or compress vectors. Yoshida & Miyato (2017) propose to add the spectral norm regularization term to the loss function of the neural network for training a robust model. Given a learning model with parameters $W$, a training dataset $S = \{(x_i, y_i)\}_{i=1}^n$, and a loss function $\mathcal{L}$. The optimization for training a robust model can be formulated as follows

$$\min_W \frac{1}{n} \sum_{i=1}^n \mathcal{L}(x_i, y_i; W) + \frac{\lambda_4}{2} \sum_{k=1}^K \sigma(M^k)^2, \tag{23}$$

where $\lambda_4$ is a regularization factor, $K$ is the layers in the neural network model, and $M^k$ is the layer-wise weight matrix for the $k$-th layer. This formulation aims to decrease the training loss as well as the spectral norms of the weight matrices. As a result, the training process encourages the model to learn weight matrices that are more stable and less sensitive to small input perturbations.

## D  THE ADOPTED MACHINE UNLEARNING METHODS

Machine unlearning aims to remove the requested data and their influence from a trained model without retraining the remaining data from scratch. In the following, we delve into more details about the adopted machine unlearning methods.

**SISA**. SISA (Bourtoule et al., 2021a) is the most representative work among existing exact data removal methods. In the SISA approach, the original training dataset $S$ is subject to a random partition into $J$ distinct shards. For each data shard $S_j$, an associated submodel $F_{S_j}(W_j)$ is trained, where $W_j \in \mathcal{W}$ represents the model parameters that parameterize the neural network model $F_{S_j}$. Subsequently, the final prediction results are derived by aggregating the outputs of the $J$ submodels using methods such as averaging or majority voting. Once an unlearning request is made, only the submodel linked to the data shard containing the requested data needs to be retrained.

**First-order Based Unlearning.** The first-order based unlearning method (Warnecke et al., 2023) leverages a first-order Taylor Series to compute the gradient updates on the model for the requested data. This method aims to find the influence of data points on the learning model by combining up-weighting and down-weighting instead of explicitly removing them. Given a pre-trained model $W^*$, a set of training data $Z = \{z_p\}_{p=1}^P \subset S$ as well as its corrected version $\tilde{Z} = \{\tilde{z}_p\}_{p=1}^P$, where $\tilde{z}_p = (x_p + \delta_p, y_p)$ and $\delta_p$ is the perturbation for the sample $x_p$. The goal is to derive a closed-form update $\Delta(Z, \tilde{Z})$ of the model parameters where $\Delta(Z, \tilde{Z})$ has the same dimensions as the learning model $W^*$ but is sparse and affects only the relevant parameters. Then, the model parameters can

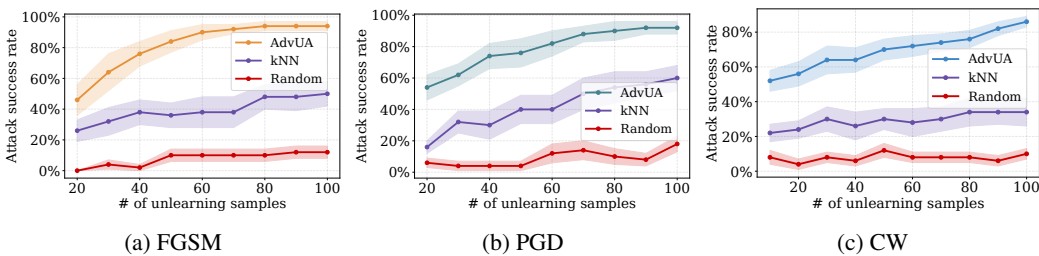

Figure 9: Attack performance of AdvUA on adversarially trained models against various attacks.

be computed by an optimal first-order update as follows

$$W^u \leftarrow W^* - \tau \big( \sum_{\tilde{z}_p \in \tilde{Z}} \nabla_W \mathcal{L}(\tilde{z}_p; W^*) - \sum_{z_p \in Z} \nabla_W \mathcal{L}(z_p; W^*) \big), \tag{24}$$

where $W^u$ is the unlearned model, $\tau$ is a pre-defined unlearning rate, and $\mathcal{L}$ is a loss function (e.g., cross-entropy). This formulation moves the pre-trained model $W^*$ towards the gradient difference between $Z$ and $\tilde{Z}$ with a step of unlearning rate $\tau$. Note that if $\tilde{Z} = \emptyset$, this method also allows model updates for removing data points, as outlined in Warnecke et al. (2023).

**Second-order Based Unlearning.** The second-order based unlearning method (Warnecke et al., 2023) uses the inverse Hessian matrix of the second-order derivatives to facilitate the unlearning process. Let $W^*$ denote a pre-trained model, $Z = \{z_p\}_{p=1}^P \subset S$ denote a set of training data, and $\tilde{Z} = \{\tilde{z}_p\}_{p=1}^P$ denote the corresponding corrected version, where $\tilde{z}_p = (x_p + \delta_p, y_p)$ and $\delta_p$ is the perturbation for the sample $x_p$. and $\delta_p$ is the unlearning modification for $x_p$. Then, the model parameters can be updated through a linear approximation as follows

$$W^u \leftarrow W^* - H_{W^*}^{-1} \big( \sum_{\tilde{z}_p \in \tilde{Z}} \nabla_W \mathcal{L}(\tilde{z}_p; W^*) - \sum_{z_p \in Z} \nabla_W \mathcal{L}(z_p; W^*) \big), \tag{25}$$

where $W^u$ is the unlearned model, $H_{W^*}^{-1}$ is the inverse Hessian matrix, and $\mathcal{L}$ is a loss function (e.g., cross-entropy). This formulation directly updates the pre-trained model $W^*$ by applying the inverse Hessian matrix $H_{W^*}^{-1}$ to the gradient difference between $Z$ and $\tilde{Z}$.

**Unrolling SGD.** Unrolling SGD unlearning method (Thudi et al., 2022) formalizes a singular gradient-based unlearning approach by expanding a sequence of stochastic gradient descent (SGD) updates using a Taylor series. In order to reverse the impact of unlearning data during the SGD training steps and obtain the unlearned model, this method involves adding the gradients of the unlearning data, computed with respect to the initial model parameters, to the final model parameters. Here, the initial model parameters can start from a pre-trained model $W^*$ and the final model parameters can be fine-tuned with $t$ SGD training steps, denoted by $W_t^*$. Let $Z = \{z_p\}_{p=1}^P \subset S$ denote a set of training data. The unlearned model can be obtained as follows

$$W^u \leftarrow W_t^* + \sum_{z_p \in Z} \frac{\eta E}{b} \nabla_W \mathcal{L}(z_p; W^*), \tag{26}$$

where $\eta$ is the learning rate, $b$ is the batch size, $E$ is the number of fine-tuning epochs (which corresponds to the number of copies of gradients presented in the SGD updates), and $\mathcal{L}$ is a loss function (e.g., cross-entropy). This formulation updates the pre-trained model $W^*$ with $t$ steps of fine-tuning $W_t^*$ by adding the gradients associated with the unlearning data as represented in the SGD training process.

# E   More Experimental Details and Results

## E.1   More Experimental Details

**Experimental Setup for Fig. 1.** In Fig. 1 of the main manuscript, we conducted preliminary experiments to investigate the adversarial robustness degradation of the undefended and defended models

Table 3: Attack transferability of AdvUA on the undefended model.

| Adversarial attack | Method | # of unlearning samples | VGG-19 | DenseNet-121 | ResNet-152 |
|---|---|---|---|---|---|
| FGSM | Baseline | None | $63.0\% \pm 4.7\%$ | $61.0\% \pm 6.9\%$ | $54.0\% \pm 4.3\%$ |
| | **AdvUA** | 10 | $\mathbf{88.0\% \pm 2.9\%}$ | $\mathbf{88.0\% \pm 2.9\%}$ | $\mathbf{82.0\% \pm 3.6\%}$ |
| | | 20 | $\mathbf{88.0\% \pm 2.9\%}$ | $\mathbf{90.0\% \pm 1.5\%}$ | $\mathbf{87.0\% \pm 3.7\%}$ |
| | | 30 | $\mathbf{92.0\% \pm 2.0\%}$ | $\mathbf{96.0\% \pm 1.6\%}$ | $\mathbf{94.0\% \pm 2.2\%}$ |
| PGD | Baseline | None | $45.0\% \pm 5.0\%$ | $49.0\% \pm 7.2\%$ | $45.0\% \pm 3.4\%$ |
| | **AdvUA** | 10 | $\mathbf{88.0\% \pm 3.3\%}$ | $\mathbf{93.0\% \pm 1.5\%}$ | $\mathbf{94.0\% \pm 2.2\%}$ |
| | | 20 | $\mathbf{91.0\% \pm 2.8\%}$ | $\mathbf{95.0\% \pm 2.2\%}$ | $\mathbf{97.0\% \pm 1.5\%}$ |
| | | 30 | $\mathbf{93.0\% \pm 2.1\%}$ | $\mathbf{97.0\% \pm 1.5\%}$ | $\mathbf{97.0\% \pm 1.5\%}$ |
| CW | Baseline | None | $48.0\% \pm 4.4\%$ | $57.0\% \pm 4.5\%$ | $43.0\% \pm 3.3\%$ |
| | **AdvUA** | 10 | $\mathbf{86.0\% \pm 3.4\%}$ | $\mathbf{94.0\% \pm 2.2\%}$ | $\mathbf{95.0\% \pm 1.7\%}$ |
| | | 20 | $\mathbf{89.0\% \pm 1.0\%}$ | $\mathbf{96.0\% \pm 1.6\%}$ | $\mathbf{96.0\% \pm 2.2\%}$ |
| | | 30 | $\mathbf{93.0\% \pm 2.1\%}$ | $\mathbf{98.0\% \pm 1.3\%}$ | $\mathbf{98.0\% \pm 1.3\%}$ |

by randomly removing some training samples in the context of machine unlearning. In the experiments, we adopt the ResNet-18 (He et al., 2016) model for the CIFAR-10 (Krizhevsky & Hinton, 2010) dataset and the ConvNet model for SVHN (Netzer et al., 2011) dataset. The undefended models are trained using standard natural algorithms, and the corresponding defended models are trained using the adversarial training technique (Madry et al., 2018) that against a PGD adversary with $l_\infty$ project gradient descent of $\epsilon = 8/255$. Subsequently, we evaluate the robustness of both undefended and defended models against FGSM (Goodfellow et al., 2015), PGD (Madry et al., 2018), and CW (Carlini & Wagner, 2017) adversarial attacks, all with a perturbation bound of $\epsilon = 8/255$. In our exploration of the adversarial vulnerability introduced by machine unlearning, it is necessary to have a comparative measure capturing the difference in adversarial robustness with and without exploiting unlearning. Therefore, we propose a general notion, referred to as the *Robust accuracy drop ratio* ($Dr_{\text{ratio}}$), to quantify this difference. The metric is defined as follows

$$Dr_{\text{ratio}} = \frac{Acc_{\text{robustness\_without\_unlearning}} - Acc_{\text{robustness\_with\_unlearning}}}{Acc_{\text{robustness\_without\_unlearning}}}, \tag{27}$$

where $Acc_{\text{robustness\_without\_unlearning}}$ denotes the robust accuracy against the adversarial attack in the absence of any unlearning process and $Acc_{\text{robustness\_with\_unlearning}}$ denotes the robust accuracy against the same adversarial attack when unlearning mechanisms are involved. Therefore, the above metric is able to reflect the degree of robustness degradation when applying machine unlearning to the undefended and defended models.

**Experimental Setup for Fig. 4.** In Fig. 4 of the main manuscript, we conducted experiments to verify the impact of unlearning on density. Specifically, we first adopt a defended model with ResNet-18 on the CIFAR-10 dataset. The defended model is trained using adversarial training against a PGD adversary with $l_\infty$ project gradient descent of $\epsilon = 8/255$. Then, we randomly select 100 hard samples and evaluate their attack success rates on the defended model against the PGD attack after unlearning. Here, we maintain a small unlearning strength (10 training samples for each hard sample) and use the first-order based unlearning method. Next, we separate the test samples into two categories: success cases (attack success after unlearning) and failure cases (attack fail after unlearning), and study their density changes before and after unlearning. More precisely, we compute the average density of success cases and failure cases within different local density scales (nearest neighbors) before and after unlearning. Our observation indicates that success cases typically exhibit higher $\ell_{wd}$ values compared to failure cases after the unlearning process, which implies that success cases are located in relatively low-density regions and are thus more susceptible to attacks.

**Parameter Settings.** In our experiments, we evaluate the efficacy of AdvUA on both undefended and defended models across a variety of datasets and model architectures. We employ a diverse set

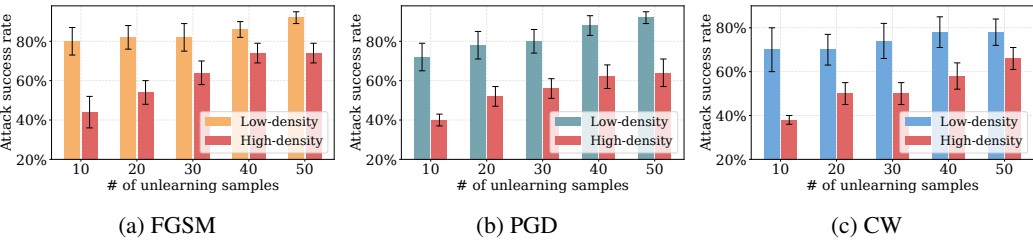

Figure 10: Impact of sampling density for AdvUA attack performance on the defended model.

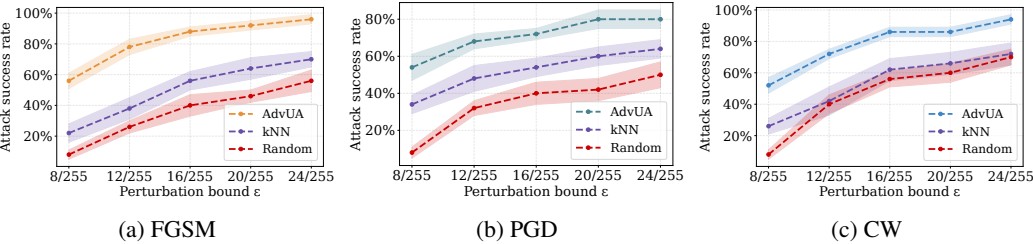

Figure 11: Impact of perturbation bound for AdvUA attack performance on the defended model.

of robust training techniques to train the defended model, and we utilize various unlearning methods to launch adversarial unlearning attacks. Specifically, for adversarial training, we train the model against a PGD adversary with $l_\infty$ project gradient descent of $\epsilon = 8/255$. When applied to the CIFAR-10 dataset, we train the model for 200 epochs with a batch size of 128. We use the SGD optimizer with an initial learning rate of 0.1 and decay the learning rate by $10\times$ at epochs 100 and 150. On the SVHN dataset, we train the model for 100 epochs with a batch size of 128. We use the SGD optimizer with a learning rate of 0.001. For interval bound propagation (Gowal et al., 2018), we train the model for 1400 epochs with a batch size of 200. The total number of training steps is 350K. We use the Adam optimizer with an initial learning rate of 0.001 and decay the learning rate by $10\times$ at steps 200K, 250K, and 300K. We set the trade-off parameter $\lambda_3 = 0.5$ in Eqn. (22). For spectral norm regularization (Yoshida & Miyato, 2017), we train the model for 100 epochs with a batch size of 200. We use the SGD optimizer with an initial learning rate of 0.01. We set the regularization factor $\lambda_4 = 0.09$ in Eqn. (23). When utilizing AdvUA to attack the undefended and defended models, we configure the trade-off parameter $\lambda_2 = 10$ in Eqn. (3). We measure the distance evaluation metric for the nearest 100 samples. Regarding the adopted unlearning methods, we set the unlearning rate to 0.0001 for the first-order based unlearning method (Warnecke et al., 2023). For the unrolling SGD unlearning method (Thudi et al., 2022), we set the fine-tuning epoch to 1 and the learning rate to 0.0001. For SISA (Bourtoule et al., 2021a), we partition the training dataset into 5 disjoint shards and attack all the shards. In the experiments of model stealing attacks, we first train the target model for 100 epochs with a batch size of 128. We use the SGD optimizer with a learning rate of 0.01. As part of the model extraction procedure, we opt to incorporate a subset of training data (10%) as unlearning samples for optimization. In this context, we adopt the first-order unlearning method with an unlearning rate of 0.01.

**Machine Configuration.** The experiments are implemented using the PyTorch (pyt, 2019) framework and run on a Linux server. This server is equipped with a GPU machine featuring AMD EPYC 32-core 2.6GHz CPUs and Nvidia A100 GPUs (40GB dedicated memory).

## E.2 MORE EXPERIMENTAL RESULTS

### E.2.1 ABOUT ATTACK PERFORMANCE AGAINST ROBUST TRAINING

In this section, we provide more experimental results about the attack performance of AdvUA on the defended models against FGSM, PGD, and CW attacks. The defended models are adversarially trained with ResNet-18 on CIFAR-10 and ConvNet on SVHN. The results for SVHN are in Fig. 9a and Fig. 9b while results for CIFAR-10 are in Fig. 9c. Here, we randomly select hard samples

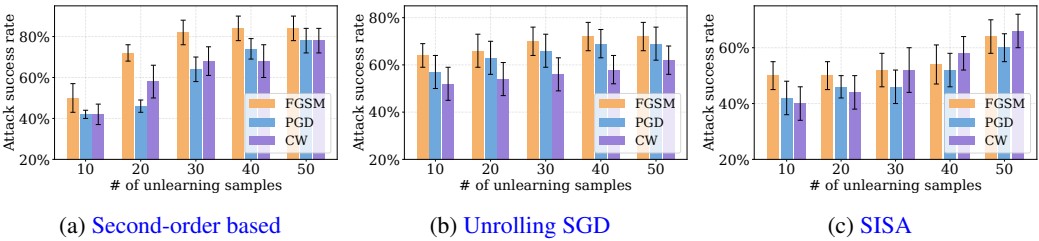

Figure 12: Attack performance of AdvUA on the defended model with various unlearning methods.

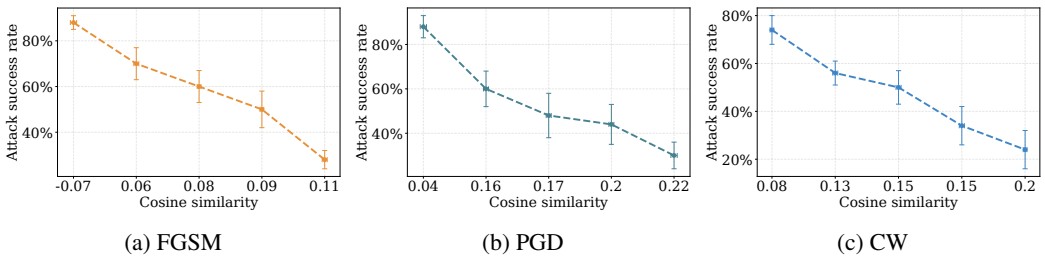

Figure 13: Correlation of attack success rate with cosine similarity (between the test sample and remaining data) on the defended model against FGSM, PGD, and CW attacks on CIFAR-10.

that are correctly classified on the defended model and then remove various quantities of training samples for each hard sample using the first-order based unlearning method. As shown in the figures, AdvUA significantly enhances the attack success rates on the defended models. For instance, when unlearning 100 training samples for each hard sample, AdvUA achieves approximately 93% on SVHN against FGSM and PGD attacks and around 86% attack success rates on CIFAR-10 against the CW attack. In contrast, the results of the random baseline and kNN baseline are limited.

### E.2.2 ABOUT EFFECTIVENESS ON ATTACK TRANSFERABILITY

In this section, more experimental results for the attack transferability with AdvUA are offered. We also use a surrogate model to unlearn various numbers of training samples and then generate the adversarial examples. Subsequently, these unlearning samples and adversarial examples are transferred to attack the target model. Table 3 presents the complete results about attack transferability of the undefended model on ImageNet, where we adopt the ResNet-50 model as the surrogate model and the VGG-19, DenseNet-121, and ResNet-152 as the target models. For comparison, we also include the attack transferability of the baseline method, which directly applies generated adversarial samples to attack the target models (without unlearning). In all attack scenarios, AdvUA outperforms the baseline by a large margin, regardless of the target model. For instance, when transferring ResNet-50 to ResNet-152 against the PGD attack, AdvUA achieves an attack success rate of 94% with 10 unlearning samples and an attack success rate of 97% with 30 unlearning samples, while the baseline only receives an attack success rate of 45%.

### E.2.3 ABOUT EXPERIMENTAL RESULTS AND DETAILS ON ABLATION STUDY

**Impact of Sampling Density.** We compare the attack success rates of hard samples sampled from low-density and high-density regions based on their $\ell_{wd}$ indicators before unlearning in Fig. 10. We also remove different numbers of unlearning samples from the defended model against FGSM, PGD, and CW attacks on CIFAR-10, using the first-order based unlearning method. As depicted in the figures, when unlearning the same number of training samples, test samples initially from a low-density region exhibit higher attack success rates than those from a high-density region. Nevertheless, even when we select the hard samples from high-density regions, AdvUA still achieves remarkable outcomes. For example, AdvUA achieves over 65% attack success rates when unlearning 50 training samples across various attacks.

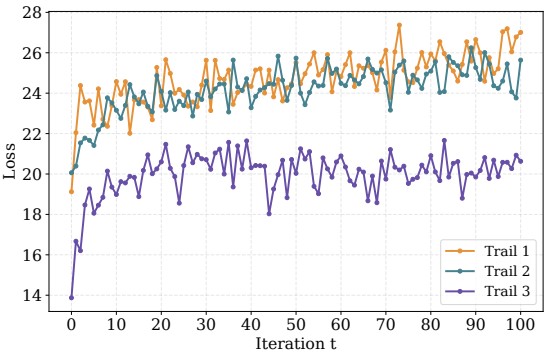

Figure 14: Optimization loss of AdvUA w.r.t number of iterations on CIFAR-10.

**Impact of Perturbation Bound.** In Fig. 11, we compare attack success rates of different perturbation bounds against FGSM, PGD, and CW attacks on CIFAR-10. We maintain a small unlearning strength (10 training samples) for each target test sample and utilize the first-order based unlearning method for removal. As we can see, the attack success rates of AdvUA increase with the increment in perturbation bound for each attack. For instance, AdvUA achieves approximately $56\%$ attack success rate with a small perturbation bound of $\epsilon = 8/255$ and about $96\%$ attack success rate with $\epsilon = 24/255$ against the FGSM attack. Despite larger perturbation bounds benefiting the baseline methods, AdvUA consistently outperforms them across various attacks.

**Attack Performance with Other Unlearning Methods.** Besides first-order based unlearning method and SISA, we also conduct experiments with other machine unlearning methods, namely second-order based and unrolling SGD, in conjunction with AdvUA. We also remove different numbers of training samples using each of the unlearning methods on the defended model against FGSM, PGD, and CW attacks on CIFAR-10. The experimental results are presented in Fig. 12. Our AdvUA exploiting these unlearning methods achieves significant attack success rates on the defended model against various attacks. For example, the second-order unlearning method attains about $80\%$ attack success rates with 50 unlearning samples. With more unlearning samples, the unlearning mechanism will remove more impact of the training samples from the defended model and further decrease the local density of the target hard samples.

### E.2.4 EXPERIMENTAL RESULTS ABOUT ADVERSARIAL DIRECTION ALIGNMENT

In the main manuscript, we propose the adversarial direction alignment to match the adversarial attack with the unlearning direction. Fig. 13 shows the correlation of attack success rate with cosine similarity on the defended model against FGSM, PGD, and CW attacks on CIFAR-10. We measure the cosine similarity between the vector of the adversarial direction and the vectors from the nearest samples to the target sample after unlearning. As shown in the figures, there is a negative correlation between the attack success rate and cosine similarity. A smaller cosine similarity corresponds to a higher attack success rate, while a larger cosine similarity corresponds to a lower attack success rate. Apparently, a small cosine similarity implies a large angle between the two vectors, indicating that the target sample is in a different direction from the nearest samples after unlearning. As a result, it tends to exhibit a higher attack success rate.

### E.2.5 EXPERIMENTAL RESULTS ABOUT OPTIMIZATION LOSS

**Convergence of the Optimization Loss.** In the main manuscript, we propose an empirical greedy approach to solve the optimization problem for the adversarial unlearning attacks, as outlined in Algorithm 1. To verify the effectiveness of our proposed approach, we examine the convergence of the optimization loss in Fig. 14. In our optimization framework, the objective is to maximize the loss of density and adversarial alignment direction. Specifically, we compute the loss for each iteration with a distance evaluation indicator and the cosine similarity between the target sample and the nearest samples in a local region after unlearning. As depicted in the figure, the loss consistently increases and eventually converges as the number of iterations increases. These results emphasize that our

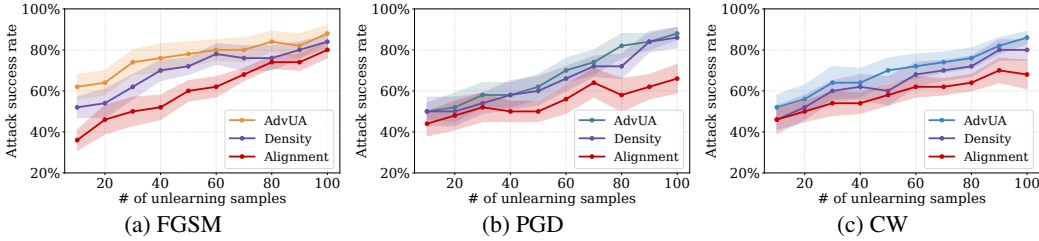

Figure 15: Attack performance of AdvUA, AdvUA with density only, and AdvUA with direction alignment only on the defended model against FGSM, PGD, and CW attacks on CIFAR-10.

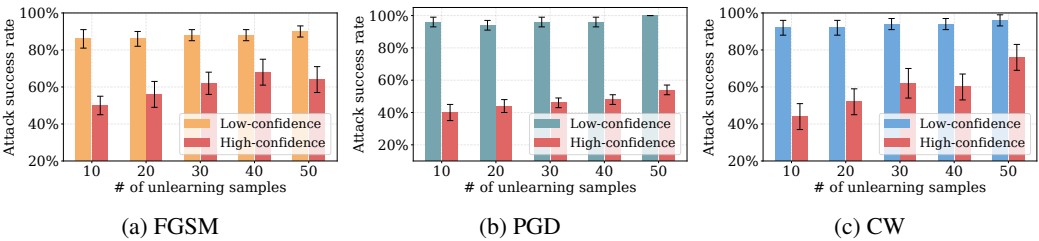

Figure 16: Attack performance of test samples sampling with low-confidence and high-confidence on the defended model against FGSM, PGD, and CW attacks on CIFAR-10.

method iteratively finds the unlearning samples that have a maximum impact on the density and alignment direction with respect to the target sample. Therefore, the experimental results provide compelling evidence to support the effectiveness of our proposed approach.

**Impact of Different Loss Components.** The overall loss of AdvUA contains two crucial components: the distance evaluation metric for density measurement and the adversarial direction for alignment assessment. Here, we conduct an ablation study to investigate the performance of these two components. Fig. 15 presents the attack performance of AdvUA, AdvUA with density only, and AdvUA with direction alignment only on the defended model against FGSM, PGD, and CW attacks on CIFAR-10. We can observe that the AdvUA incorporating both components outperforms the AdvUA that solely takes density into account or solely focuses on direction alignment. In particular, the density measurement appears to play a more dominant role in guiding the target sample in a low-density region, and direction alignment assists in pulling the representation of the target sample towards the low-density region.

### E.2.6 EXPERIMENTAL RESULTS ABOUT SAMPLES WITH DIFFERENT CONFIDENCE

In Fig. 16, we illustrate the attack performance of test samples sampling from low-confidence and high-confidence on the defended model against FGSM, PGD, and CW attacks on CIFAR-10. We express the confidence of the test sample by the probability value obtained after applying the softmax function. Specifically, we randomly select hard samples with a probability less than 0.8 as representing low confidence and those with a probability greater than 0.8 as indicating high confidence. As shown in the figures, when unlearning the same number of training samples, test samples with low confidence exhibit higher attack success rates than those with high confidence. For instance, in the case of the FGSM attack, when we unlearn 50 training samples, test samples with low confidence achieve an attack success rate of approximately $90\%$, while those with high confidence achieve an attack success rate of around $64\%$. These experimental results align with the observations presented in Fig. 2. Low-confidence test samples tend to be covered by fewer training samples and are not well-trained in the model, so they are usually located in low-density regions, while high-confidence test samples are usually located in high-density regions.

### E.2.7 EXPERIMENTAL RESULTS ABOUT GENERALIZATION IN ATTACKING ROBUST VISION TRANSFORMERS (VITS)

In this section, we conduct experiments to evaluate the generalization ability of the proposed AdvUA in attacking robust ViTs (Mo et al., 2022; Peng et al., 2023; Dosovitskiy et al., 2020). Note that the key idea behind ViTs (Dosovitskiy et al., 2020) is to apply the Transformer architecture to visual data, and the Transformer architecture relies on self-attention mechanisms, enabling it to capture relationships and dependencies between different elements (e.g., regions of an image) in a sequence of data. In the experiments, we adopt the vit_base_patch16_224_in21k model in (Mo et al., 2022) and train an adversarial robust model on 20% of the CIFAR-10 dataset using the adversarial training technique. Notably, to train the adversarial robust model, we follow (Mo et al., 2022; Peng et al., 2023) to utilize adversarial training to improve the model's robustness against adversarial attacks. In the adversarial training process, we train robust ViTs against a PGD adversary with an $l_\infty$ perturbation budget of $\epsilon = 8/255$. Then we conduct our AdvUA attack on this adversarially robust model against PGD attacks, and our experimental results show that AdvUA can effectively attack robust ViTs. For example, when unlearning 50 unlearning samples, AdvUA can achieve an attack success rate of $56.0\% \pm 4.0\%$ while the kNN baseline achieves $30.0\% \pm 6.8\%$; when unlearning 100 unlearning samples, AdvUA can achieve an attack success rate of $68.0\% \pm 6.1\%$ while the kNN baseline achieves $42.0\% \pm 6.3\%$. Note that in the experiments, we choose the hard samples to attack, which led to a zero attack success rate on the original robust model before unlearning. Therefore, these experiments further underscore the significance of the proposed AdvUA, and help us to well understand adversarial unlearning attacks in different settings.

