# OpenReview forum: "Rethinking Adversarial Robustness in the Context of the Right to be Forgotten"
_ICLR.cc/2024/Conference — Submitted to ICLR 2024_

### Official Review · Reviewer_dMF7 · 2023-10-20

**Soundness:** 3 good
**Presentation:** 3 good
**Contribution:** 4 excellent
**Rating:** 6
**Confidence:** 3

**Summary:**

The authors introduce and highlight the notion that adversarially robust models can become more vulnerable to adversarial attacks after the model performs machine unlearning on a specified set of training examples. In light of this insight, the authors propose Adversarial Unlearning Attack (AdvUA), an unlearning attack that significantly reduces the robustness of the target model. Specifically, AdvUA selects training examples to unlearn that are both close to the target victim examples and in the same directional alignment as the adversarial attack. Experiments on 4 datasets using 6 different models, 3 adversarial attacks, and 2 unlearning techniques suggest AdvUA selects training examples to unlearn that lead to significantly less robust models than the chosen baseline methods: a kNN based approach that selects the k-nearest training examples to unlearn, and a random method which selects training to remove uniformly at random.

**Strengths:**

* The authors highlight an important vulnerability of adversarially robust models, this work is especially pertinent given the rapid rise in popularity of different machine unlearning techniques.

* The proposed method, AdvUA, is a simple approach that uses regional density and adversarial attack direction to maliciously select training examples to unlearn. The simplicity and effectiveness of this approach makes it potentially generalizable to a wide range of models and domains.

* The paper is relatively well-written and easy to follow.

* Experiments with different models, datasets, adversarial attacks, and unlearning techniques provides evidence that AdvUA outperforms the chosen baseline selection methods.

* The authors also evaluate AdvUA in black-box settings related to attack transferability and model stealing attacks.

**Weaknesses:**

* One of my main concerns is the robustness of the results. The experimental setup section describes the experiments use 4 datasets, 6 model architectures, 3 adversarial attacks, and 2 unlearning techniques. This setup enables more than 140 different dataset-model-attack-unlearning combinations, however, the results in each section tend to only show a very select subset of these combinations. For example, Figure 6 only shows 2 attacks on the CIFAR-10 dataset, and 1 attack for the SVHN dataset; why aren't all the results shown for all the dataset-model-attack combinations, or aggregated in some way to give readers a general sense of AdvUA's effectiveness across different settings?

* I think there could be a more in-depth discussion between the $k$NN based approach and AdvUA using local density only. A comparison of $k$NN to this ablated version of AdvUA may be insightful and beneficial to readers. It's also not clear to me how $k$ is chosen in the experiments, and how impactful the choice of $k$ is on the results.

* It's not clear to me how robust AdvUA is to different unlearning techniques. I think a plot comparing results using the two different unlearning techniques would provide more evidence to the generalizability of the proposed approach. Have the authors also experimented with the gold standard unlearning technique of retraining from scratch on a small dataset/model?

* The clarity of the paper could be improved, please see the "Questions" section for details.

* There are some minor grammatical errors throughout the paper, consider using a service like Grammarly to fix these issues.

* Consider listing what datasets and models are being shown in Figure 6.

* Figures 1, 7, and 8 are not colorblind friendly.

**Questions:**

* What is the non-robust accuracy of the different models in Figure 1? It may be beneficial to readers to see that comparison.

* Is Figure 2 showing the density of training examples before or after unlearning?

* What dataset is being shown Figure 4? Also, how significant is the $\ell_{wd}$ gap between "before" and "after" unlearning for each local density scale in Figure 4? What does the "local density scale" represent, and how does it differ from $\ell_{wd}$?

* Where are the kNN and random baseline methods in Figure 7?

* Have the authors thought about how AdvUA can be used to train more robust models?

---

> ### Author Response · Authors · 2023-11-18
> **Response to Reviewer dMF7 (Part 1/4)**
>
> Thank you for your valuable comments and suggestions, and they are very helpful for us to improve our paper. We have carefully integrated them in the revised version. Our point-by-point responses to your comments are provided below. In the final version, we will include all of them and credit anonymous reviewers in the acknowledgments section.
>
> **Q4.1. Discussing the robustness of experimental results across 140 different dataset-model-attack-unlearning combinations in experiments (since there are 4 datasets, 6 model architectures, 3 adversarial attacks, and 2 unlearning techniques).**
>
> **A4.1:** Thanks for the valuable comment. We want to first clarify that in the initial submission, we adopted 4 unlearning techniques instead of 2 unlearning techniques. Due to space limitation, we deferred the experimental results on another two unlearning methods to the Appendix (see Figure 12 in the Appendix with detailed discussion in Section E.2.3.).
>
> Additionally, we appreciate the suggestion to exhaustively test every possible scenario across all of the 140 different dataset-model-attack-unlearning combinations in experiments. Considering the limited rebuttal time, we conducted the following new attack transferability experiments to show the good attack transferability of AdvUA across various settings (e.g., different unlearning methods). Specifically, in Table 1, we illustrate the attack transferability of AdvUA across different unlearning methods. From this table, we can see that the malicious unlearning samples generated using the first-order based unlearning method are also effective in increasing the models’ adversarial vulnerabilities when applied with other unlearning methods. The reason is that different unlearning methods are designed to achieve similar unlearning performance to remove the impact of the given same set of unlearning samples, and the unlearning samples we found are crucial against the robust training. Note that in Table 1, we adopted the ResNet-18 model, the adversarial attack method of FGSM, and the CIFAR-10 dataset. Additionally, in Table 2 and Table 3, we conducted experiments to verify the attack transferability of AdvUA across different models under our black-box settings. Specifically, in Table 2, AdvUA demonstrates good attack transferability across different models when transferring unlearning samples. Here, we adopted the first-order based unlearning method and the adversarial attack method of FGSM. Moving to Table 3, AdvUA also exhibits effective attack transferability across different models when transferring adversarial samples. More specifically, we first obtained different unlearned models by transferring the same set of unlearning samples to different models using the first-order based unlearning method, and then transferred adversarial samples across these obtained unlearned models for PGD adversarial attacks.
>
> Table 1: Transferability of AdvUA across different unlearning methods. The first-order based unlearning method serves as the surrogate unlearning method.
> | # of unlearning samples | Second-order based | Unrolling SGD |
> | :--- | :--- | :--- |
> | 20 | 52.0% ± 4.4% | 58.0% ± 6.3% |
> | 40 | 60.0% ± 6.7% | 60.0% ± 6.0% |
> | 60 | 78.0% ± 8.1% | 60.0% ± 6.7% |
> | 80 | 86.0% ± 3.1% | 66.0% ± 6.7% |
> | 100 | 88.0% ± 4.4% | 70.0% ± 4.5% |
>
> Table 2:  Attack transferability (transferring unlearning samples) across different models. ResNet-50 serves as the surrogate model.
> | # of unlearning samples | VGG-19 | DenseNet-121 | ResNet-152 |
> | :--- | :--- | :--- | :--- |
> | 10 | 88.0% ± 2.9% | 91.0% ± 4.3% | 85.0% ± 3.4% |
> | 20 | 93.0% ± 2.6% | 93.0% ± 3.3% | 88.0% ± 4.2% |
> | 30 | 94.0% ± 2.7% | 96.0% ± 2.2% | 93.0% ± 3.3% |
>
> Table 3: Attack transferability (transferring adversarial samples) across different models. ResNet-50 serves as the surrogate model.
> | # of unlearning samples | VGG-19 | DenseNet-121 | ResNet-152 |
> | :--- | :--- | :--- | :--- |
> | 10 | 88.0% ± 3.3% | 93.0% ± 1.5% | 94.0% ± 2.2% |
> | 20 | 91.0% ± 2.8% | 95.0% ± 2.2% | 97.0% ± 1.5% |
> | 30 | 93.0% ± 2.1% | 97.0% ± 1.5% | 97.0% ± 1.5% |
>
> In the final version, we will also incorporate this helpful suggestion. By conducting the above attack transferability across various settings (e.g., varying unlearning methods), we believe that they can provide a good sense of the attack generalization performance of our proposed method across diverse scenarios.
>
> **Q4.2: Why Figure 6 only shows 2 attacks on the CIFAR-10 dataset, and 1 attack for the SVHN dataset?**
>
> **A4.2:** Thanks for the question. We want to clarify that in our initial submission, we have also conducted experiments to investigate the attack performance of AdvUA on the defended models against CW attacks on the CIFAR-10 dataset, and verify the attack performance of AdvUA on the defended models against FGSM and PGD attacks on the SVHN dataset. Due to space limitations, these experimental results are deferred to Figure 9 in the Appendix.

---

> ### Author Response · Authors · 2023-11-18
> **Response to Reviewer dMF7 (Part 2/4)**
>
> **Q4.3. Giving a more in-depth discussion between the kNN based approach and AdvUA using local density only. A comparison of kNN to this ablated version of AdvUA may be insightful and beneficial to readers. It's also not clear to me how $k$ is chosen in the experiments, and how impactful the choice of $k$ is on the results.**
>
> **A4.3:** Thanks for the valuable comments. The kNN method, relying on absolute distances, removes the k-nearest points around the victim sample, which limits its ability to accurately assess local data structures. In contrast, our local density measure $l_{wd}$ for AdvUA, as defined in Equation (1),  leverages relative distances and takes into account broader data areas. This allows for a more precise characterization of local structures and is more effective in identifying crucial points that are key to maintaining model performance near victim samples.
>
> Additionally, following your suggestions, in Table 4 below, we conducted experiments to compare the attack success rates of the kNN method and our AdvUA approach using local density only, where $k$ equals the number of unlearning samples for fair comparison. This comparison is conducted across various numbers of unlearning samples with FGSM on the CIFAR-10 dataset. These experimental results verify that the AdvUA method with local density only, outperforms the kNN approach under different values of $k$. In addition, we can find that increasing the value of $k$ in the kNN method does not significantly enhance attack performance. In contrast, our approach substantially improves attack performance as the number of unlearning samples increases. This is attributed to the effectiveness of our method in effectively reducing local density by removing more crucial training samples.
>
> Table 4: Attack success rate comparison between AdvUA with local density and kNN.
> | # of unlearning samples    | 10   | 20   | 30   | 40   | 50   |
> | :--- | :--- | :--- | :--- | :--- | :--- |
> | kNN baseline | 32.0% ±  8.0% | 30.0% ± 7.5% | 34.0% ± 9.5% | 32.0% ± 9.5% | 30.0% ± 10.4% |
> | AdvUA with local density | 52.0% ±  5.3% | 54.0% ± 6.7% | 62.0% ± 6.3% | 70.0% ± 5.4% | 72.0% ± 4.4%|
>
> Lastly, as explained above, in our experiments, $k$ is set as the number of unlearning samples for fair comparison with AdvUA. Note that in our experiments, we also set the attack budget for the launched adversarial unlearning attacks by restricting the maximal number of unlearning samples to be deleted.
>
> **Q4.4. How robust AdvUA is to different unlearning techniques? Giving a plot to compare results using the 2 different unlearning techniques to show the generalizability of the proposed method.**
>
> **A4.4:** Thanks for highlighting this. Due to space limitations in our main submission, we deferred the experiments about the comparison of AdvUA's performance across various unlearning methods in the Appendix. Detailed experimental results are available in Figure 12, along with a comprehensive discussion in Section E.2.3. For your convenience, we have also provided the following table (Table 5) to show the effectiveness of AdvUA with second-order based, unrolling SGD, and SISA unlearning methods to attack the defended model against FGSM attacks on CIFAR-10.
>
> Table 5: Attack success rate of AdvUA with different unlearning methods.
> | # of unlearning samples | 10 | 20 | 30 | 40 | 50 |
> | :--- | :--- | :--- | :--- | :--- | :--- |
> | Second-order based | 50.0% ± 6.8% | 72.0% ± 4.4% | 82.0% ± 5.5% | 84.0% ± 5.8% | 84.0% ± 5.8% |
> | Unrolling SGD | 64.0% ± 5.0% | 66.0% ± 7.3% | 70.0% ± 6.1% | 72.0% ± 6.1% | 72.0% ± 6.1% |
> | SISA | 50.0% ± 5.4% | 50.0% ± 5.4% | 52.0% ± 6.1% | 54.0% ± 6.7% | 64.0% ± 5.8% |
>
>
>
> **Q4.5. Conducting new experiments with the gold standard unlearning technique of retraining from scratch on a small dataset/model.**
>
> **A4.5:** Thanks for the constructive comment. Following your suggestions, we have conducted new experiments to evaluate AdvUA on the unlearning method of retraining from scratch. Table 6 presents the experimental results obtained by applying the ConvNet model to 20% of the SVHN dataset and testing against FGSM attacks. From this table, we can easily see that AdvUA can also achieve good attack performance when unlearning samples through retraining from scratch.
>
> Table 6: Attack success rate of AdvUA with retraining.
> | # of unlearning samples | 20 | 40 | 60 | 80 | 100 |
> | :--- | :--- | :--- | :--- | :--- | :--- |
> | AdvUA | 50.0% ± 8.0% | 52.0% ± 7.4% | 56.0% ± 6.5% | 60.0% ± 8.9% | 68.0% ± 4.4% |
> | kNN | 26.0% ± 6.7% | 24.0% ± 5.0% | 26.0% ± 5.2% | 26.0% ± 4.3% | 30.0% ± 8.0% |
> | Random | 24.0% ± 5.0% | 24.0% ± 6.5% | 24.0% ± 5.8% | 22.0% ± 7.0% | 26.0% ± 6.7% |
>
> **Q4.6. Correcting minor grammatical errors.**
>
> **A4.6:** Thanks for the helpful suggestion. Following your suggestions, in the revised version, we have thoroughly checked the paper and carefully corrected grammatical errors.

---

> ### Author Response · Authors · 2023-11-18
> **Response to Reviewer dMF7 (Part 3/4)**
>
> **Q4.7. Listing what datasets and models are being shown in Figure 6.**
>
> **A4.7:** Thanks for the comment. The details of Figure 6 are provided at the top of Page 8 in the main manuscript. Specifically, Figure 6a and Figure 6b utilize the ResNet-18 model on the CIFAR-10 dataset, and Figure 6c employs ConvNet on the SVHN dataset.
>
> **Q4.8. Changing Figures 1, 7, and 8 to be colorblind friendly.**
>
> **A4.8:** Thanks for the valuable suggestions. Following your suggestions, in the revised version, we have revised Figures 1, 7, and 8 to make them colorblind friendly.
>
> **Q4.9. What is the non-robust accuracy of the different models in Figure 1?**
>
> **A4.9:** Thanks for bringing attention to the non-robust accuracy in Figure 1. As you suggested, we have conducted experiments to evaluate the non-robust accuracy of the different models in Figure 1. Table 7 below reports the non-robust accuracy on the SVHN dataset, which is the test accuracy of the naturally trained model under adversarial attacks (e.g., FGSM, PGD, and CW attacks). Specifically, we randomly removed 10% of the training samples and re-evaluated the non-robust accuracy of the unlearned model. As observed in this table, the random deletion strategy adversely affects the non-robust accuracy against various adversarial attacks. For example, the non-robust accuracy of the FGSM attack is 22.62% before unlearning and drops to 9.56% after unlearning. In other words, even random unlearning can also exacerbate the adversarial vulnerabilities of the undefended models.
>
> Table 7: Comparison of non-robust accuracy on SVHN.
> | Adversarial attack | Before unlearning | After unlearning |
> | :--- | :--- | :--- |
> | FGSM | 22.62% ± 0.00% | 9.56% ± 0.27% |
> | PGD | 3.75% ± 0.01% | 1.37% ± 0.12% |
> | CW | 1.13% ± 0.01% | 0.32% ± 0.05% |
>
> **Q4.10. Is Figure 2 showing the density of training examples before or after unlearning?**
>
> **A4.10:** To clarify, Figure 2 shows the density of training examples after unlearning, as detailed in Section 3.2 FORMULATION OF ADVUA.
>
> **Q4.11. What dataset is being shown in Figure 4? Also, how significant is the $\ell_{wd}$ gap between “before” and “after” unlearning for each local density scale in Figure 4? What does the “local density scale” represent, and how does it differ from $\ell_{wd}$?**
>
> **A4.11:** Thanks for the insightful questions. We want to first clarify that in Figure 4, we adopted the CIFAR-10 dataset, and the detailed experimental setup for Figure 4 is deferred to Appendix Section E.1 due to space limitation.
>
> Additionally, in Figure 4, the $\ell_{wd}$ gap between “before” and “after” unlearning for a specific local density scale $K_{v}$ reflects the change in the concentration of nearest data samples around a victim sample “before” and “after” unlearning. Note that the $\ell_{wd}$ metric is designed to characterize the local behavior around a specific point. From Figure 4b, we can easily see that when the local density scale is set as 50, the ratio of the $\ell_{wd}$ value after unlearning to the $\ell_{wd}$ value before unlearning is around 120%. Notably, the higher $\ell_{wd}$ means the victim samples are placed in a lower density region, which means a greater likelihood of these samples being susceptible to attack. To give more intuition about how significant the $\ell_{wd}$ gap between “before” and “after” unlearning, in Table 8 below, we give the attack success rate (ASR) “after” unlearning under different local density scales. Notably, “before” unlearning, the attack success rate for all cases in Table 8 is zero. From Table 8, we can see that “after” unlearning, success cases (where we can successfully attack the target victim samples and ASR is 100%) typically have higher $\ell_{wd}$ compared to failure cases (where we cannot successfully attack the target victim samples and ASR is 0%). This highlights the significance of increasing $\ell_{wd}$ (“after” unlearning) through the unlearning of malicious samples to facilitate successful attacks. Note that inspired by this, our proposed method is designed in a way to increase $\ell_{wd}$ via deliberately unlearning some specific unlearning samples to successfully launch adversarial unlearning attacks.
>
> Table 8: Attack success rate under different local density scales.
> | Local density scale | 50 | 100 | 150 | 200 |
> | :--- | :--- | :--- | :--- | :--- |
> | ASR $(\ell\^{before}\_{wd}\/ \ell\^{after}\_{wd})$ of success cases | 100% (26.08/30.26) | 100% (24.93/27.78) | 100% (23.26/25.42) | 100% (22.6/24.33) |
> | ASR $(\ell\^{before}\_{wd}\/ \ell\^{after}\_{wd})$ of failure cases | 0% (20.93/24.64) | 0% (19.5/22.12) | 0% (18.77/20.54) | 0% (18.15/19.69) |
>
> Lastly, to clarify, the local density scale refers to the number of nearest neighboring data points that are taken into account when calculating the density metric $\ell_{wd}$. The detailed explanations for the local density scale can be found in the third paragraph in Section 3.2 of the main manuscript.

---

> ### Author Response · Authors · 2023-11-18
> **Response to Reviewer dMF7 (Part 4/4)**
>
> **Q4.12. Where are the kNN and random baseline methods in Figure 7?**
>
> **A4.12:** Thanks for the questions. As you suggested, following the same experimental setup in Figure 7, we have conducted new experiments on the kNN and Random baseline methods, and presented the obtained experimental results in Table 9 and Table 10 below. Specifically, Table 9 shows the obtained experimental results of AdvUA and the suggested baselines on the certified defended models using spectral normal regularization against FGSM attacks. Table 10 shows the obtained experimental results of AdvUA and the suggested baselines on the certified defended models using interval bound propagation against PGD attacks. It is evident that AdvUA outperforms the kNN baseline and Random baseline across varying unlearning samples, demonstrating the efficacy of our proposed method against certified defenses.
>
> Table 9: Attack success rate of AdvUA on certified defended models using spectral norm regularization.
> | # of unlearning samples | 20 | 40 | 60 | 80 | 100 | 120 | 140 | 160 | 180 | 200 |
> | :--- | :--- | :--- | :--- | :--- | --- | :--- | :--- | :--- | :--- | :--- |
> | AdvUA | 42.0% ± 4.7% | 48.0% ± 5.3% | 46.0% ± 7.9% | 52.0% ± 5.3% | 52.0% ± 6.1% | 60.0% ± 7.9% | 68.0% ± 4.4% | 72.0% ± 6.8% | 70.0% ± 6.8% | 76.0% ± 5.8% |
> | kNN | 38.0% ± 4.7% | 22.0% ± 7.0% | 42.0% ± 8.1% | 30.0% ± 6.8% | 40.0% ± 8.4% | 36.0% ± 6.5% | 40.0% ± 8.4% | 40.0% ± 8.3% | 44.0% ± 8.3% | 52.0% ± 7.4% |
> | Random | 26.0% ± 7.9% | 22.0% ± 9.6% | 28.0% ± 4.4% | 26.0% ± 3.1% | 34.0% ± 7.9% | 34.0% ± 5.2% | 42.0% ± 7.6% | 42.0% ± 11.3% | 44.0% ± 7.2% | 46.0% ± 5.2% |
>
> Table 10: Attack success rate of AdvUA on certified defended models using interval bound propagation.
> | # of unlearning samples | 20 | 40 | 60 | 80 | 100 | 120 | 140 | 160 | 180 | 200 |
> | :--- | :--- | :--- | :--- | :--- | --- | :--- | :--- | :--- | :--- | :--- |
> | AdvUA | 38.0% ± 7.6% | 44.0% ± 9.3% | 48.0% ± 6.8% | 50.0% ± 6.1% | 50.0% ± 6.8% | 60.0% ± 3.0% | 64.0% ± 5.0% | 76.0% ± 5.8% | 74.0% ± 4.3% | 80.0% ± 5.2% |
> | kNN | 28.0% ± 6.1% | 38.0% ± 7.6% | 30.0% ± 7.5% | 46.0% ± 6.0% | 32.0% ± 9.0% | 40.0% ± 6.0% | 32.0% ± 6.8% | 52.0% ± 6.1% | 48.0% ± 5.3% | 60.0% ± 6.7% |
> | Random | 28.0% ± 7.4% | 34.0% ± 6.7% | 42.0% ± 3.6% | 40.0% ± 4.2% | 44.0% ± 7.2% | 34.0% ± 5.2% | 40.0% ± 6.7% | 42.0% ± 3.6% | 42.0% ± 6.3% | 52.0% ± 4.4% |
>
>
> **Q4.13. Have the authors thought about how AdvUA can be used to train more robust models?**
>
> **A4.13:** Thanks for raising the discussion on using AdvUA to train more robust models. Indeed, AdvUA can also be employed to enhance model robustness by targeting the local density regions during the training process. It's important to note that in our paper, AdvUA is originally designed to unlearn certain training samples to strategically place target victim samples in low-density areas. By considering this process in reverse, AdvUA can be adapted to target the low-density areas during the training process to enhance model robustness. Specifically, this can be achieved by combining AdvUA with current data augmentation techniques to produce synthetic augmented data that targets low-density regions at different training stages. Notably, this strategy is complementary to existing adversarial robust training methods. It can be readily combined with these methods to develop enhanced adversarial robust training strategies, which is also observed from our experiments.

---

> ### Author Response · Authors · 2023-11-22
> **Message from authors**
>
> Dear Reviewer dMF7,
>
> Given that the author/reviewer discussion phase concludes in just one day, we would like to inquire again if our responses have satisfactorily addressed your concerns and questions.
>
> We take every comment seriously and have provided our point-by-point responses as well as the revised version. Please let us know if you have any further concerns so that we can reply promptly till the end of the discussion phase (Nov. 22).
>
> Thank you very much for your time and efforts for reviewing our work.

---

> > ### Comment · Reviewer_dMF7 · 2023-11-22
> >
> > I thank the authors for their detailed response and have increased my score as a result.

---

> > > ### Author Response · Authors · 2023-11-22
> > > **Thank you!**
> > >
> > > Dear Reviewer dMF7,
> > >
> > > Thank you for raising your score and for your constructive comments. We are glad that our responses have addressed your concerns!
> > >
> > > Thank you again for the time and effort you have dedicated to reviewing our work!

---

### Official Review · Reviewer_aqii · 2023-10-30

**Soundness:** 3 good
**Presentation:** 3 good
**Contribution:** 3 good
**Rating:** 6
**Confidence:** 3

**Summary:**

Based on the hypothesis that unlearning increases the vulnerability of deep learning models to adversarial attacks, this study examines methods to achieve adversarial attacks using unlearning on models without using adversarial perturbations. The authors have experimentally demonstrated that unlearning with randomly selected samples increases vulnerability to adversarial attacks. Also,
 the authors introduced an algorithm that increases vulnerability more efficiently by selecting samples to be unlearned based on the density of the training data and the direction of the target sample.

**Strengths:**

The relationship between unlearning and adversarial robustness has never been discussed before, and this study makes us aware of a new research direction for the community. The experiments are meticulously conducted, and the results are reliable.

**Weaknesses:**

The intent of Theorem 2 is not clear. Could you please explain what theoretical justification this theorem gives about the proposed method?

**Questions:**

Question:
The intent of Theorem 2 is not clear. Could you please explain what theoretical justification this theorem gives about the proposed method?

Comments:
- Unlearning U should be defined as a mapping with specifying inputs and output.
- H should be should be defined before it appears for the first time.

---

> ### Author Response · Authors · 2023-11-18
> **Response to Reviewer aqii**
>
> Thank you for your valuable comments and suggestions, and they are very helpful for us to improve our paper. We have carefully integrated them in the revised version. Our point-by-point responses to your comments are provided below. In the final version, we will include all of them and credit anonymous reviewers in the acknowledgments section.
>
> **Q3.1. The intent of Theorem 2.**
>
> **A3.1:** Thanks for the constructive comment. Theorem 2 illustrates the relation between the number of training samples and the associated $l_{\infty}^{\epsilon}$-adversarial robust error. This theorem shows that when the training set size diminishes from $n_1$ to $n_2$ under the conditions mentioned in this theorem, there is a high probability (at least $1-2 \exp \left(-\frac{d}{8(\sigma^2+1)}\right)$) that the robust error will increase. Note that our proposed AdvUA is designed to unlearn the strategically selected malicious samples to reduce the models’ adversarial robustness. In the revised version, we have provided more explanations for the intent of this theorem in the paragraph following Theorem 2 in Page 6.
>
> **Q3.2. Giving the mapping definition of Unlearning $U$ via specifying inputs and output.**
>
> **A3.2:** We appreciate the helpful suggestion. Following your suggestions, in the revised version (please refer to Definition 1 on Page 3), we have defined Unlearning $U$ by specifying inputs and output.
>
> **Q3.3. Defining $H$ before it appears for the first time.**
>
> **A3.3:** Thanks for pointing this out. Following your suggestions, in the revised version (please refer to the first paragraph in Section 2), we have first defined $H$ before it appears for the first time.

---

> > ### Comment · Reviewer_aqii · 2023-11-21
> > **Response to the rebuttal**
> >
> > Thanks for the rebuttal. The responses were satisfactory to me.

---

> > > ### Author Response · Authors · 2023-11-21
> > > **Thank you!**
> > >
> > > Dear Reviewer aqii,
> > >
> > > We appreciate your positive feedback. We are glad that our responses have addressed your concerns!
> > >
> > > Thank you again for the time and effort you have dedicated to reviewing our work!

---

### Official Review · Reviewer_eRuv · 2023-10-31

**Soundness:** 3 good
**Presentation:** 2 fair
**Contribution:** 3 good
**Rating:** 6
**Confidence:** 3

**Summary:**

In contrast to previous works that is focused on the adversarial robustness of a simple classifier, this paper takes care of a new setting: how to decrease the robustness of models during theunlearning process. Correspondingly, they proposed a new attack which is called AdvUA. The effectivenss of AdvUA is evaluated in both white-box and black-box setting. In addition,  AdvUA can further improve the performance of model stealing attack.

**Strengths:**

1 The soundness of AdvUA is good.

2 The experiment is relatively sufficient

3 It seems that AdvUA will bring significant gain to ASR of the mainstream attack, e.g. C&W and PGD attack.

**Weaknesses:**

1 Although the experimental section shows promising gain on ASR after applying AdvUA, an important metric is obviously ignored: **the benign accuracy** of the model after unlearning. Ideally, a powerful attack will not only improve the ASR  but also has marginal impact on the benign function of the threat model. Therefore, I would suggest authors report both ACC  and ASR instead of ASR alone.

2 Some minor error exists: For example,"PDG" in Page 7.  I would suggest authors re-read the whole paper for further corrections.

3 The writing of this paper is not easy to follow for me. Some partitions of paper are redundant: For example, after definition 2, "the attacker misleads the unlearned model $W^u$ to directly misclassify the target victim samples without any further perturbations." What insights does this conclusion provide for the design of AdvUA?

4 The computational cost of AdvUA could be unaffordable when the dataset is huge. Algorithm 1 shows that AdvUA needs to at least iterate through the data $|S_e|\times B\times |K_v|$ times. When $|S_e|$ is huge, large $|K_v|$ is needed to ensure the accuracy of accessment.  It may sharply increase the computational expenses of AdvUA.

**Questions:**

1 Why don't the starting points of AdvUA, kNN, and Random overlap in Figure 6?

2 Almost all experiments are performed on CNNs. ViT are the another prevailing architecture in computer vision. Can AdvUA help exisiting attacks better envade robust ViTs [1,2] during unlearning?

3 The authors only consider one kind of black-box settings:  The attacker use a surrogate model to unlearn various numbers of training samples with the first-order based method and then generate the adversarial examples. However, another black-box setting is that:  The attackers use a surrogate model to select samples for unlearning and victims use the selected samples to unlearn their own models. Is AdvUA effective in this situation?

For other questions, please refer to the weakness section.

[1] Mo et al. When Adversarial Training Meets Vision Transformers: Recipes from Training to Architecture. In NeurIPS 2022

[2] Peng et al., RobArch: Designing Robust Architectures against Adversarial Attacks, In arXiv 2023

---

> ### Author Response · Authors · 2023-11-18
> **Response to Reviewer eRuv (Part 1/3)**
>
> Thank you for your valuable comments and suggestions, and they are very helpful for us to improve our paper. We have carefully integrated them into the revised version. Our point-by-point responses to your comments are provided below. In the final version, we will include all of them and credit anonymous reviewers in the acknowledgments section.
>
> **Q2.1. Giving the benign accuracy of the model after unlearning.**
>
> **A2.1:** Thanks for the constructive suggestion. Following your suggestion, in Table 1 below, we report the attack success rate and the benign accuracy of the models after unlearning. We followed the experimental setup in Figure 6a that uses ResNet-18 on CIFAR-10 and against FGSM attacks. It's worth noting that the initial model achieved a benign accuracy of 83.53%. From this table, we can see that AdvUA not only achieves high attack success rates but also demonstrates a minor impact on the benign accuracy of the threat model.
>
> Table 1: Benign accuracy and attack success rate of AdvUA.
> | # of unlearning samples | 10 | 20 | 30 | 40 | 50 |
> | :--- | :--- | :--- | :--- | :--- | :--- |
> | Attack success rate | 62.0% ± 6.3% | 64.0% ± 5.8% | 74.0% ± 6.0% | 76.0% ± 6.5% | 78.0% ± 5.5% |
> | Benign accuracy | 81.52% ± 0.27% | 80.86% ± 0.29% | 80.22% ± 0.33% | 79.33% ± 0.42% | 78.68% ± 0.37% |
>
> **Q2.2. Correcting some minor error (e.g., “PDG” in Page 7) and re-reading the paper for further corrections.**
>
> **A2.2:** Thanks for the helpful suggestions. Following your suggestions, in the revised version, we have corrected the mentioned minor error, and thoroughly re-read the whole paper for further corrections.
>
> **Q2.3. Some partitions of paper are redundant for the paper writing, e.g., what insights does “the attacker misleads the unlearned model $W^{u}$ to directly misclassify the target victim samples without any further perturbations.” provide for AdvUA?**
>
> **A2.3:** Thanks for the constructive suggestion. We want to first clarify that adversarial robustness refers to the ability of a model to maintain its performance in the presence of adversarial perturbations. For the statement “the attacker misleads the unlearned model $W^{u}$ to directly misclassify the target victim samples without any further perturbations”, we want to emphasize the fact that simply unlearning certain specific training samples can suffice to cause the direct misclassification of targeted victim samples, without the need for any additional adversarial perturbations. This poses a greater threat compared to where misclassifications occur as a result of adversarial attacks after unlearning. Following your suggestions, in the final version, we will carefully review and resolve the redundant partitions in the paper.

---

> ### Author Response · Authors · 2023-11-18
> **Response to Reviewer eRuv (Part 2/3)**
>
> **Q2.4. Discussions on the computational cost of AdvUA in Algorithm 1.**
>
> **A2.4:** Thanks for the valuable comment. Note that due to the discrete deletion nature of unlearning samples and the corresponding time complexity of combinatorial optimization, solving the proposed loss in Equation (3) in the main manuscript is computationally infeasible. To address this, we instead adopt the empirical search based method (as outlined in Algorithm 1) to effectively optimize the loss. First, we would argue that from the aspects of practical attack constraints and attack stealthiness, the computational cost of AdvUA is relatively low and affordable. In practice, using a large number of malicious unlearning samples (i.e., $B$) could alert the model owner to potential malicious attack risks, thereby increasing the likelihood of detecting the launched attacks. Therefore, in Algorithm 1, we set the maximum number of malicious unlearning samples as $B$. Additionally, the attacker in practice usually has a limited search set $S_{e}$, and consequently, the number of the top nearest neighbors (i.e., $K_{v}$) is also limited. In our experiments, we also typically opted for small values  for $|S_{e}|$, $K_{v}$ ($K_{v} < |S_{e}|$), and the malicious unlearning samples. For example, as depicted in Figure 6, we set $|S_{e}|$ as 500, and with a maximum of 40 unlearning samples, we achieved a 76% attack success rate. Note that in all of the experiments in our paper, we set $K_{v}$ as 100. We also want to emphasize that the attack performance being constrained by a limited budget is also of great practical significance in existing literature.
>
> Additionally, we want to clarify that in our experimental attack implementation of AdvUA in the initial submission, when attacking the victim samples, we did not unlearn each unlearning sample individually. Instead, in our experiments, we first split all the available unlearning samples from the overall search set (i.e., $S_{e}$) into a set of local search subsets. Then, during our empirical search based optimization procedure, at each step, we selected an unlearning search subset, whose removal will largely increase the loss value associated with the distance evaluation metric and cosine similarity. In the revised version, we have revised Algorithm 1 in the Appendix to include this detail. We also conducted new experiments to compare the effectiveness of searching through the entire pool (global search) versus a local pool (local search), in terms of both running time and attack success rate. In Table 2 and Table 3 below, we presented the obtained experimental results. Specifically, we set the candidate set sizes (i.e., $|S_{e}|$) to 100, 250, and 500, with the objective of unlearning a total of $B=50$ samples in each pool. In the global search, our method involves iterating over each sample in the candidate set to identify the effective malicious unlearning samples. On the other hand, in local search, we split the set $S_{e}$ into multiple subsets based on the above procedure. From the below table, we can see that our adopted strategy has less running time while achieving comparable attack performance.
>
> Table 2: Running time (min) of AdvUA with global search and local search.
> | $\|S_e\|$ | 100 | 250 | 500 |
> | :--- | :--- | :--- | :--- |
> | Global search | 4.05 ± 0.10 | 11.91 ± 0.31 | 25.30 ± 0.53 |
> | Local search | 0.29 ± 0.00 | 0.87 ± 0.0 | 1.66 ± 0.04 |
>
> Table 3: Attack success rate of AdvUA with global search and local search.
> | $\|S_e\|$ | 100 | 250 | 500 |
> | :--- | :--- | :--- | :--- |
> | Global search | 56.0% ± 7.2% | 60.0% ± 9.4% | 64.0% ± 6.5% |
> | Local search | 58.0% ± 8.7% | 56.0% ± 6.5% | 62.0% ± 6.3% |
>
> **Q2.5. Why don’t the starting points of AdvUA, kNN, and Random overlap in Figure 6?**
>
> **A2.5:** Thanks for the comment. To clarify, the starting point for the number of unlearning samples in Figure 6 is 10 for Figure 6a & Figure 6b (CIFAR-10), and 20 for Figure 6c (SVHN), rather than zero, explaining why AdvUA, kNN, and Random don't overlap. In the revised version, we have corrected this, shifting all the starting points to zero. Note that the samples we choose to attack are all hard samples, which cannot be successfully attacked on the original robust model before unlearning. Thus, in this figure, the attack success rate when no samples are unlearned is just zero.

---

> ### Author Response · Authors · 2023-11-18
> **Response to Reviewer eRuv (Part 3/3)**
>
> **Q2.6. Can AdvUA help existing attacks better invade robust ViTs [1, 2] during unlearning?**
>
> **A2.6:** Thank you very much for bringing our attention to the application of our proposed AdvUA to invade robust ViTs [1, 2]. Following your suggestions, we have conducted experiments to evaluate the attack performance of AdvUA on robust ViTs [1]. We adopted the vit_base_patch16_224_in21k model in [1] and trained an adversarial robust model on 20% of the CIFAR-10 dataset using the adversarial training technique. We then conducted our AdvUA attack on this robust model against PGD attacks, and our experimental results show that AdvUA can effectively attack robust ViTs. For example, when unlearning 50 unlearning samples, AdvUA can achieve an attack success rate of 56.0% ± 4.0% while the kNN baseline achieves 30.0% ± 6.8%; when unlearning 100 unlearning samples, AdvUA can achieve an attack success rate of 68.0% ± 6.1% while the kNN baseline achieves 42.0% ± 6.3%. Note that in this experiment, we chose the hard samples to attack, which led to a zero attack success rate on the original robust model before unlearning. Therefore, these experiments further underscore the significance of AdvUA, and help us to well understand adversarial unlearning attacks in different settings.
>
> **Q2.7. Conducting experiments on another black-box setting where the attackers use a surrogate model to select samples for unlearning and victims use the selected samples to unlearn their own models.**
>
> **A2.7:** Thanks for highlighting another black-box setting. Following your suggestions, in Table 4 below, we evaluate the attack performance of AdvUA against FGSM in another black-box setting. Specifically, we first crafted malicious unlearning samples by utilizing the ResNet-50 model, and then transferred these crafted unlearning samples to the black-box victim models (i.e., VGG-19, DenseNet-121, and ResNet-152). After that, we unlearned these crafted samples from the black-box victim models (i.e., VGG-19, DenseNet-121, and ResNet-152), resulting in the corresponding unlearned models, which were then used to generate adversarial perturbations. As shown in the table below (i.e., Table 4), AdvUA can also achieve high attack performance in this suggested black-box setting.
>
> Table 4: Attack success rate of AdvUA in another black-box setting. ResNet-50 serves as the surrogate model.
> | # of unlearning samples | VGG-19 | DenseNet-121 | ResNet-152 |
> | :--- | :--- | :--- | :--- |
> | 10 | 88.0% ± 2.9% | 91.0% ± 4.3% | 85.0% ± 3.4% |
> | 20 | 93.0% ± 2.6% | 93.0% ± 3.3% | 88.0% ± 4.2% |
> | 30 | 94.0% ± 2.7% | 96.0% ± 2.2% | 93.0% ± 3.3% |
>
> **Reference**
>
> [1] Yichuan Mo, Dongxian Wu, Yifei Wang, Yiwen Guo, and Yisen Wang. When adversarial training meets vision transformers: Recipes from training to architecture. Advances in Neural Information Processing Systems, 35:18599–18611, 2022.
>
> [2] ShengYun Peng, Weilin Xu, Cory Cornelius, Kevin Li, Rahul Duggal, Duen Horng Chau, and Jason Martin. Robarch: Designing robust architectures against adversarial attacks. arXiv preprint arXiv:2301.03110, 2023.

---

> ### Author Response · Authors · 2023-11-22
> **Message from authors**
>
> Dear Reviewer eRuv,
>
> Given that the author/reviewer discussion phase concludes in just one day, we would like to inquire again if our responses have satisfactorily addressed your concerns and questions.
>
> We take every comment seriously and have provided our point-by-point responses as well as the revised version. Please let us know if you have any further concerns so that we can reply promptly till the end of the discussion phase (Nov. 22).
>
> Thank you very much for your time and efforts for reviewing our work.

---

> > ### Comment · Reviewer_eRuv · 2023-11-22
> > **Thank you for your rebuttal**
> >
> > Dear authors,
> >
> > I have carefully read the rebuttal. Considering ViTs as prevailing architectures, I would suggest adding the experiments of ViTs to the paper to demonstrate the generalization ability of your proposed AdvUA.
> >
> > Look forward to your reply.
> >
> > Best,
> >
> > Reviewer eRuv

---

> > > ### Author Response · Authors · 2023-11-22
> > > **Thank you for the prompt response**
> > >
> > > Dear Reviewer eRuv,
> > >
> > > We really appreciate your prompt response, and the helpful suggestions that help us improve our paper. We totally agree with you about the importance of demonstrating the generalization ability of AdvUA to attack ViTs. Following your suggestions, in the latest revised version (please refer to Section E.2.7 in Page 25), we have added the experiments of ViTs. These experiments effectively demonstrate AdvUA's generalization performance in attacking ViTs, further underscoring its practical significance in the field. In the final version, we will also incorporate all of your suggestions.
> > >
> > > Thank you once again for your time and constructive input.

---

> > > > ### Comment · Reviewer_eRuv · 2023-11-22
> > > > **Thank you for the update**
> > > >
> > > > Dear authors,
> > > >
> > > > I have raised my score. At last, I recommend to use \citep instead of \cite. It seems that the citation in Appendix E.2.7 does not follow the instruction of ICLR24. If you have enough time, I would recommend update a new version.
> > > >
> > > > Best regards,
> > > >
> > > > Reviewer eRuv

---

> > > > > ### Author Response · Authors · 2023-11-22
> > > > > **Thank you**
> > > > >
> > > > > Dear Reviewer eRuv,
> > > > >
> > > > > Thank you for raising your score and for your prompt helpful responses. We have corrected the citation in Appendix E.2.7 in the latest revised version.
> > > > >
> > > > > Thank you so much for devoting time to improving our work!

---

### Official Review · Reviewer_HikF · 2023-10-31

**Soundness:** 2 fair
**Presentation:** 3 good
**Contribution:** 3 good
**Rating:** 5
**Confidence:** 4

**Summary:**

Unlearning allows machine learning models to forget training data, however existing machine unlearning methods do not consider the security implications of unlearning. The authors propose a new attack called Adversarial Unlearning Attack (AdvUA), which can exploit a security vulnerability in machine unlearning to reduce the adversarial robustness of unlearned models (white and black box scenario). AdvUA learns the decision boundary information of the victim target model and it works by generating malicious unlearning requests that target specific parts of the model that are important for decision making. This makes it easier for the attacker to extract decision boundary information by observing how the model behaves on the remaining training data.

**Strengths:**

The paper makes a significant contribution to the field of machine learning security by highlighting the importance of considering security when designing and implementing machine unlearning methods.  Strengths:
- Well written paper and identifies an important security vulnerability in machine unlearning.
- Proposes an effective attack, AdvUA, that can exploit vulnerability to significantly reduce the adversarial robustness of unlearned models.
- Provides a comprehensive evaluation of AdvUA on a variety of models and datasets, demonstrating its effectiveness against both standard and certified defense methods.

**Weaknesses:**

1) AdvUA can be computationally expensive, especially for large and complex machine learning models. How does the computation time complexity of AdvUA against certified defense methods vary based on the number of unlearning samples?

2) The authors should conduct further evaluation on real-world systems to better understand the practical impact of AdvUA.

**Questions:**

Questions for authors:

Concern 1: Results show that AdvUA can perform reasonably well against certified defense methods, even when unlearning only a small number of training samples. How does the effectiveness of AdvUA against certified defense methods vary depending on the specific certified defense method used?

Concern 2: It is about the AdvUA attack’s computational time complexity in black box scenarios. How does the computation time complexity of AdvUA against certified defense methods vary based on the number of unlearning samples?

Concern 3: The effectiveness of AdvUA depends on the selection of the malicious unlearning requests, it is possible to design an algorithm to select suboptimal set of malicious unlearning requests?

---

> ### Author Response · Authors · 2023-11-18
> **Response to Reviewer HikF (Part 1/3)**
>
> Thank you for your valuable comments and suggestions, and they are very helpful for us to improve our paper. We have carefully integrated them into the revised version. Our point-by-point responses to your comments are provided below. In the final version, we will include all of them and credit anonymous reviewers in the acknowledgments section.
>
> **Q1.1. Discussions on the time complexity, e.g., how does the computation time complexity of AdvUA against certified defense methods vary based on the number of unlearning samples?**
>
> **A1.1:** Thanks for the constructive comment. We want to first clarify that AdvUA is built upon existing machine unlearning methods, which are designed to efficiently unlearn the requested samples to be deleted from the well-trained (large and complex) models when compared with retraining from scratch. It should be noted that these well-trained models can be trained by using certified defense techniques, and these defense processes are implemented and completed before the unlearning process begins. For example, the time complexity of the proposed first-order based unlearning method in [1] is independent of the presence or type of defense methods used in training the original models. On the other hand, due to the processing required for searching each potential malicious unlearning request from the original model, the time complexity tends to increase when the number of unlearning samples increases.
>
> Additionally, our AduvUA is designed to identify $B$ effective malicious unlearning samples from the available search set $S_{e}$ based on existing unlearning methods. For AdvUA, we would argue that from the aspects of practical attack constraints and the need for attack stealthiness, its search time complexity (for searching effective malicious unlearning samples) is relatively low and affordable. The reason is that a larger number of malicious unlearning samples would increase the attacker’s risks of being detected, and the attacker in practice typically has limited data. Based on these concerns, in our experiments, we typically opted for small values when setting the size of the available search set (i.e., $|S_{e}|$) and the total number of malicious unlearning samples, which work well in our experiments. For example, as depicted in Figure 6 in our paper, we set the size of the search set (i.e., $|S_{e}|$) as 500, and we can achieve an attack success rate of approximately 76% when the number of malicious unlearning samples is limited to a maximum of 40. We also want to emphasize that the attack performance being constrained by a limited budget is also of great practical significance in existing literature.
>
> Further, we want to clarify that in our experimental attack implementation, when targeting the victim samples, we did not unlearn each unlearning sample individually. Instead, in our experiments, we first split all of the available unlearning samples from the overall search set (i.e., $|S_{e}|$) into a set of local search subsets. Then, during our empirical search-based optimization procedure, at each step, we selected an unlearning search subset, whose removal will largely increase the loss value associated with the distance evaluation metric and cosine similarity. Note that due to discrete deletion formulations of unlearning samples and the corresponding complexity of combinatorial optimization, solving the proposed loss in Equation (3) in the main manuscript is computationally infeasible. To address this, we instead adopt the empirical search based method. In the revised version, we have revised Algorithm 1 in the Appendix to clarify our experimental attack implementation in our experiments.
>
> Lastly, following your suggestions, we also conducted experiments about the impact of the number of unlearning samples on the computation complexity. In Table 1 below, we present the attack success rate and the computation cost per target victim sample. We performed AdvUA against FGSM attack on CIFAR-10 with ResNet-18 and made the candidate set $|S_{e}|=500$. As shown in this table, the computational cost increases with the number of unlearning samples, and more computational costs correspond to higher attack success rates. Nevertheless, the overall computational cost remains reasonable in our experiments.
>
> Table 1: Attack success rate and computation cost of AdvUA.
> | # of unlearning samples | 10 | 20 | 30 | 40 | 50 |
> | :--- | :--- | :--- | :--- | :--- | :--- |
> | Attack success rate | 62.0% ± 6.3% | 64.0% ± 5.8% | 74.0% ± 6.0% | 76.0% ± 6.5% | 78.0% ± 5.5% |
> | Computation cost (min) | 1.40 ± 0.01 | 2.56 ± 0.03 | 3.50 ± 0.05 | 4.49 ± 0.21 | 5.04 ± 0.06 |

---

> ### Author Response · Authors · 2023-11-18
> **Response to Reviewer HikF (Part 2/3)**
>
> **Q1.2. Evaluation on real-world systems.**
>
> **A1.2:** Thanks for the valuable comment. Following your suggestions, we have conducted new experiments by considering real-world settings. However, due to the constraints (e.g., ethical and legal considerations) of attacking real-world running systems, we followed [2] and used a real-world medical dataset to simulate a healthcare diagnosis system, a domain that commonly integrates deep learning models to enhance diagnostics. Building upon this simulation, we extended the method in [2] by employing AdvUA on medical deep learning based systems. Specifically, we adopted the dermoscopy image dataset, a large collection of multi-source dermatoscopic images, for building an automated diagnosis system. Next, we trained a robust ResNet-18 model using the adversarial training technique on this dataset. Then, we conducted our AdvUA method on the defended model against PGD attacks, and the obtained experimental results also verify the effectiveness of AdvUA. For example, when unlearning 100 training samples, AdvUA can achieve an attack success rate of 86.0% ± 3.1%, while the kNN baseline achieves 22.0% ± 7.6%. Therefore, these conducted experiments further highlight the practical importance and impact of AdvUA on real-world system applications.
>
> **Q1.3. How does the effectiveness of AdvUA against certified defense methods vary depending on the specific certified defense method used?**
>
> **A1.3:** Thanks for the insightful question. In our experiments, we have indeed observed variations in attack effectiveness against certified defense methods. Specifically, in our conducted experiments, we found that if the adopted certified defense methods (e.g., interval bound propagation-based certified defenses adopted in our paper) modify the training data to provide certified guarantees, they are highly susceptible to malicious unlearning samples. On the other hand, for the certified defense methods that rely on upper bound-based regularizers (e.g., spectral norm based methods adopted in our experiments) to provide their guarantees, we observed that under the same number of malicious unlearning samples, these methods typically show less vulnerability to security threats when subjected to malicious unlearning samples.

---

> ### Author Response · Authors · 2023-11-18
> **Response to Reviewer HikF (Part 3/3)**
>
> **Q1.4. In the black-box setting, e.g., how does the computation time complexity of AdvUA against certified defense methods vary based on the number of unlearning samples?**
>
> **A1.4:** Thanks for the valuable question. Note that in the main manuscript, for the black-box setting discussed in Page 6, we assume that the adversary only knows the output of the target model through predictions. In this black-box setting, we can first follow existing model stealing attacks to construct a surrogate model, which leverages shared decision boundaries among various models. Then, based on this constructed surrogate model, we can perform our proposed adversarial unlearning attacks (i.e., AdvUA) to attack the target victim samples. After that, we can transfer the crafted malicious unlearning samples (that are generated based on the constructed surrogate model) to attack the victim black-box model, which means that the computation cost of AdvUA in terms of the number of unlearning samples in this black-box setting is similar to that in the white-box setting (as discussed earlier in Q1.2). Notably, in this black-box setting, the additional computational cost is incurred in the construction of the surrogate model. This step is essential for the black-box attack to approximate the behavior of the target victim model, thereby enabling the effective transfer of malicious unlearning samples.
>
> **Q1.5. Is it possible to design an algorithm to select suboptimal set of malicious unlearning requests?**
>
> **A1.5:** Thanks for the valuable comment. Yes, we can enhance the performance AdvUA by first selecting a suboptimal candidate set as follows: Firstly, given the whole search set $S_{e}$ and the target victim samples $\\{x_{v}\\}_{v=1}^{V}$, we can utilize existing influence function techniques [3] at the initialization step to estimate the influence of each sample in $S_e$ on the victim target samples. Next, for each victim sample $x_v$, we can identify a subset within $S_e$ containing samples that have a large influence on the prediction of $x_v$. Then, we merge these identified search subsets to form a suboptimal set. Based on this, when attacking the victim target samples during the optimization procedure, we can craft effective malicious unlearning samples from this suboptimal search set (instead of the original whole search set $S_e$). The reason behind this strategy is that targeted sample deletions in this selected influential suboptimal set are more likely to impact the areas highly relevant to the victim target samples.
>
> **Reference**
>
> [1] Alexander Warnecke, Lukas Pirch, Christian Wressnegger, and Konrad Rieck. Machine unlearning of features and labels. Network and Distributed System Security Symposium, 2023.
>
> [2] Samuel G Finlayson, Hyung Won Chung, Isaac S Kohane, and Andrew L Beam. Adversarial attacks against medical deep learning systems. arXiv preprint arXiv:1804.05296, 2018.
>
> [3] Pang Wei Koh and Percy Liang. Understanding black-box predictions via influence functions. In International conference on machine learning, pp. 1885–1894. PMLR, 2017.

---

> ### Author Response · Authors · 2023-11-22
> **Message from authors**
>
> Dear Reviewer HikF
>
> Given that the author/reviewer discussion phase concludes in just one day, we would like to inquire again if our responses have satisfactorily addressed your concerns and questions.
>
> We take every comment seriously and have provided our point-by-point responses as well as the revised version. Please let us know if you have any further concerns so that we can reply promptly till the end of the discussion phase (Nov. 22).
>
> Thank you very much for your time and efforts for reviewing our work.

---

> ### Author Response · Authors · 2023-11-23
> **A gentle reminder**
>
> Dear Reviewer HikF,
>
> As the discussion period approaches its end, we would greatly appreciate it if you could confirm whether our responses have adequately addressed your concerns. Please feel free to share any additional questions or concerns you may have. We are ready to provide prompt responses until the conclusion of the discussion phase.
>
> We sincerely appreciate the time and effort you've invested in reviewing our work and look forward to your feedback.

---

### Author Response · Authors · 2023-11-18
**Global Response to the Reviewers**

Dear Reviewers:

We would like to first express our great gratitude to all the reviewers for their time and valuable comments. We highly appreciate all the feedback and suggestions, which further help us improve our paper. We are greatly encouraged that they found our ideas and contributions to be novel and significant (Reviewers HikF, eRuv, aqii, and dMF7), technically solid (Reviewers HikF, eRuv, and dMF7), and effective (Reviewers HikF, eRuv, aqii, and dMF7). We are grateful for the positive comments regarding the writing quality and clarity of our paper (Reviewers HikF and dMF7). We are thankful that they recognized the effectiveness of our proposed method (Reviewers HikF, eRuv, aqii, and dMF7), and the comprehensiveness of our conducted experiments (Reviewers HikF, eRuv, and aqii)

Additionally, we have provided point-by-point responses for the reviewers. Although we have provided individual responses for the reviewers, we wanted to summarize what has been pointed out in the reviews, for which we have provided the corresponding responses.

* For **Reviewer HikF**, the raised comments that we have addressed are given below.
    * Regarding the analysis of computational cost, we have provided the analysis and the running time in A1.1 and A1.4.
    * Regarding evaluations on real-world systems, we have followed your suggestion and the obtained experimental results can be found in A1.2.
    * Regarding discussions on the effectiveness of AdvUA on the specific adopted certified defense methods and the selection of the suboptimal set, we have provided our individual responses in A1.3 and A1.5.

* For **Reviewer eRuv**, the raised comments that we have addressed are given below.
    * Regarding conducting new experiments on the benign accuracy, robust ViTs, and another black-box setting, we have conducted these suggested experiments (please refer to A2.1, A2.6, and A2.7).
    * Regarding further clarifications about the computational cost and the starting points in Figure 6, we have provided further clarifications in A2.4 and A2.5.
    * Regarding minor errors and redundant paper partitions for paper writing, we have provided our respective responses in A2.2 and A2.3.

* For **Reviewer aqii**, the raised comments that we have addressed are given below.
    * Regarding explaining the intent of Theorem 2, we have provided more explanations about the intent of Theorem 2 in A3.1.
    * Regarding definitions of Unlearning $U$ and $H$, we have provided the responses in A3.2 and A3.3.

* For **Reviewer dMF7**, the raised comments that we have addressed are given below.
    * Regarding experiments across 140 different dataset-model-attack-unlearning combinations and Figure 6, our detailed responses can be found in A4.1 and A4.2.
    * Regarding giving discussions between the kNN based method and AdvUA using local density only and the choice of $k$, we have provided the detailed responses in A4.3.
    * Regarding discussing the robustness of AdvUA across different unlearning techniques and conducting new experiments on retraining from scratch, our individual responses are given in A4.4 and A4.5.
    * Regarding grammatical errors and listing the datasets (and models) adopted in Figure 6, our responses can be found in A4.6 and A4.7.
    * Regarding changing figure colors and providing clarifications about the raised five questions in the “Questions” section, we have provided the responses in A4.8, A4.9, A4.10, A4.11, A4.12, and A4.13.

Further, based on the reviewer’s comments, in the uploaded revised version, we have made the corresponding changes, which are highlighted in blue, to the initially submitted paper.

We look forward to participating in further discussions and answering any further questions that the reviewers may have.

Thank you very much for your time and consideration.

Best regards,

Authors of Paper3049

---

### Author Response · Authors · 2023-11-21
**Message from authors**

Dear Reviewers,

We would like to express our sincerest gratitude for the time and effort you have dedicated to reviewing our work.

As we are presently in the discussion phase, we would appreciate it if you let us know what your remaining concerns are with our work before this discuss phase ends. Could you kindly let us know whether our recent responses and revisions have addressed your questions and concerns effectively? We hope that these changes have provided a clear understanding of our contribution. Please rest assured that we are fully prepared to address any additional concerns that may arise.

We eagerly look forward to hearing from you at your earliest convenience.

Thank you once again for your time and valuable input.

---

### Meta-Review · Area_Chair_fFZx · 2023-12-08

**Metareview:**

The paper presents a novel and significant contribution to the field of machine learning security, particularly in the context of unlearning and its implications on adversarial robustness. However, there are concerns about its computational efficiency, the breadth of its experimental validation, and the clarity of some of its theoretical aspects.

Computational Efficiency: Adv UA is noted to be potentially computationally expensive, especially in black box scenarios. This raises questions about its practical applicability in real-world systems, which is a crucial aspect for the acceptance of such research.

Experimental Evaluation: There is a noted lack of comprehensive experimental evaluation. While the paper provides some experimental results, they are not sufficiently broad to confidently assess the effectiveness and robustness of Adv UA across different scenarios. This limitation is particularly concerning given the complexity and diversity of real-world applications.

Theoretical Clarity: The intent of certain theoretical aspects, like Theorem 2, is not clearly explained, leading to uncertainties about the foundational principles underpinning the proposed method.

Impact on Benign Accuracy: The paper does not adequately address the impact of Adv UA on the benign accuracy of models post-unlearning. This is a significant oversight as it directly relates to the practicality of the proposed method in a real-world context.

Writing and Presentation: Reviewers have pointed out issues with the clarity of writing and presentation. Redundancies and minor grammatical errors detract from the overall quality of the manuscript.

Robustness of Results: Concerns were raised about the robustness of the results presented. The experimental setup described allows for a wide range of combinations, yet the results shown are limited to a select subset. This lack of comprehensive presentation of results limits the ability to fully assess the efficacy of Adv UA.

Thus, the paper does not meet the high standards required for acceptance.

**Justification For Why Not Higher Score:**

As said in meta review.

**Justification For Why Not Lower Score:**

NA

---

### Decision · Program_Chairs · 2024-01-16

Reject